# The glycosphingolipid MacCer promotes synaptic bouton formation in Drosophila by interacting with Wnt

Yan Huang[1], Sheng Huang[1,2†], Coralie Di Scala[3], Qifu Wang[1], Hans H Wandall[4], Jacques Fantini[5], Yong Q Zhang[1]*

[1]Key Laboratory for Molecular and Developmental Biology, Institute of Genetics and Developmental Biology, CAS Center for Excellence in Brain Science and Intelligence Technology, Beijing, China; [2]Sino-Danish College, Sino-Danish Center for Education and Research, Chinese Academy of Sciences, Beijing, China; [3]INMED U1249, INSERM, Marseille, France; [4]Copenhagen Center for Glycomics, Department of Cellular and Molecular Medicine, Faculty of Health Sciences, University of Copenhagen, Copenhagen, Denmark; [5]UNIS UMR_S 1072, INSERM, Aix-Marseille Université, Marseille, France

*For correspondence:
yqzhang@genetics.ac.cn

Present address: †Institute of Biology, Freie Universität Berlin, Berlin, Germany

Competing interests: The authors declare that no competing interests exist.

**Abstract** Lipids are structural components of cellular membranes and signaling molecules that are widely involved in development and diseases, but the underlying molecular mechanisms are poorly understood, partly because of the vast variety of lipid species and complexity of synthetic and turnover pathways. From a genetic screen, we identify that mannosyl glucosylceramide (MacCer), a species of glycosphingolipid (GSL), promotes synaptic bouton formation at the *Drosophila* neuromuscular junction (NMJ). Pharmacological and genetic analysis shows that the NMJ growth-promoting effect of MacCer depends on normal lipid rafts, which are known to be composed of sphingolipids, sterols and select proteins. MacCer positively regulates the synaptic level of Wnt1/Wingless (Wg) and facilitates presynaptic Wg signaling, whose activity is raft-dependent. Furthermore, a functional GSL-binding motif in Wg exhibiting a high affinity for MacCer is required for normal NMJ growth. These findings reveal a novel mechanism whereby the GSL MacCer promotes synaptic bouton formation via Wg signaling.
DOI: https://doi.org/10.7554/eLife.38183.001

## Introduction

The glycosphingolipids (GSLs) are particularly abundant in the nervous system and are essential for brain development (*Fantini and Yahi, 2015*; *Yu et al., 2009*). Specific deletion of GSLs in mouse brain leads to severe neural defects or lethality (*Jennemann et al., 2005*). Nevertheless, the molecular and cellular functions of GSLs in development are poorly understood, partially due to the complexity of GSL metabolism and the variety of GSL structures in vertebrates. Because of the comparatively simple metabolic pathways and the power of genetic studies in invertebrates (*Bellen and Yamamoto, 2015*; *Zhu and Han, 2014*), a mechanistic understanding of the role of GSLs in development and function of the nervous system is beginning to be established (*Dahlgaard et al., 2012*; *Huang et al., 2016*; *Kniazeva et al., 2015*; *Yonamine et al., 2011*).

GSLs and sphingomyelins (SMs, another class of sphingolipids) assemble with sterol into detergent-resistant membrane microdomains known as lipid rafts, which are critical for signal transduction and membrane trafficking (*Lingwood and Simons, 2010*). In neural development, multiple processes such as neuronal proliferation, recognition, migration, and synapse formation are regulated by lipid rafts (*Aureli et al., 2015*). For instance, disruption of rafts by depletion of sphingolipid or sterol

leads to enlargement and gradually loss of synapses in cultured hippocampal neurons (*Hering et al., 2003*). Many growth factors or their receptors are preferentially located within and functionally dependent on membrane rafts (*Wang and Yu, 2013*; *Watanabe et al., 2009*; *Zhai et al., 2004*). Specifically, GSLs interact with raft-associated signaling proteins, such as epidermal growth factor receptor (EGFR) and Notch, thereby facilitating signal transduction (*Coskun et al., 2011*; *Hamel et al., 2010*; *Wang and Yu, 2013*). Our previous study uncovered that the GSL mannosyl glucosylceramide (MacCer) promotes NMJ overgrowth (*Huang et al., 2016*). However, the mechanisms by which GSLs mediate in vivo neural development remain elusive.

Normal brain function depends on proper formation of synaptic connections. The *Drosophila* larval glutamatergic neuromuscular junction (NMJ) is an advantageous model for dissecting mechanisms underlying synaptic development (*Bayat et al., 2011*; *Khuong et al., 2013*; *Khuong et al., 2010*; *Korkut and Budnik, 2009*). To uncover potential functions of lipids at synapses, we used the *Drosophila* NMJ as a model synapse and performed a genetic screen targeting genes involved in lipid biosynthesis and turnover pathways. From this screen, we identified multiple genes involved in sphingolipid de novo synthesis affecting NMJ development. We further found that MacCer is both required and sufficient for promoting NMJ growth and bouton formation in presynaptic neurons. MacCer promotes NMJ growth in a raft-dependent manner. We revealed that MacCer positively regulates synaptic Wg level and the presynaptic activity of Wg signaling. Further multiple independent assays showed MacCer physically interacts with Wg via a previously unidentified GSL-binding motif in Wg. Mutations in this motif disrupt the MacCer-Wg binding and normal NMJ growth. These findings demonstrate that the GSL MacCer plays a crucial role in bouton formation and NMJ growth and uncover a novel regulatory mechanism of Wg signaling pathway by MacCer.

## Results

### Mutations in de novo sphingolipid synthetic enzymes affect NMJ growth

To gain novel insights into the role of lipids in regulating synaptic development, we carried out a genetic screen targeting genes involved in the biosynthesis and turnover of fatty acids, glycerophospholipids, and sphingolipids. We tested over 60 candidate genes by examining NMJ morphology (*Supplementary file 1*) and identified two enzymes, serine palmitoyltransferase 2 Lace and ceramide synthase Schlank, promoting NMJ bouton formation as mutations in either of the two proteins led to fewer and larger boutons (*Figure 1B–F,H,I,K*). Mutations in *lace* and *schlank* disrupt the de novo synthesis of ceramides, the central intermediate in sphingolipid synthesis/metabolism (*Figure 1A*; *Adachi-Yamada et al., 1999*; *Bauer et al., 2009*; *Fyrst et al., 2004*). These data indicate that depletion in de novo synthesis of ceramides inhibits bouton formation. In addition to the de novo ceramide synthesis, the ceramide precursor sphingosines can be phosphorylated by sphingosine kinase 2 (Sk2) to produce phosphorylated sphingosines (*Figure 1A*). In *Sk2* mutants, the level of phosphorylated sphingosines is reduced, while sphingosines accumulate (*Fyrst et al., 2004*; *Yonamine et al., 2011*). We found that mutations in *Sk2* resulted in more satellite boutons at NMJs (*Figure 1G,J*), in contrast to the fewer and larger bouton phenotype in *lace* and *schlank* mutants. These results indicate that the de novo synthesis of ceramides, their downstream derivatives, or both promote bouton formation and NMJ growth.

GSL and ceramide phosphoethanolamine (CerPE, the *Drosophila* analogue of SM) are major membrane sphingolipids modified from ceramides (*Figure 1A*). However, we did not study the effects of CerPE in NMJ growth because there are multiple genes encoding candidates of CerPE synthase in the *Drosophila* genome (*Vacaru et al., 2013*; *Supplementary file 1*).

### GSL synthases Egh and Brn bi-directionally regulate NMJ growth presynaptically

In contrast to the complex CerPE synthesis, the first three steps of GSL synthetic pathway are catalyzed by single GSL synthases in *Drosophila* (*Figure 2A*; *Chen et al., 2007*; *Schwientek et al., 2002*; *Wandall et al., 2003*). Disruption of the first one glucosylceramide synthase (GlcT1) to block GSL biosynthesis by feeding larvae with a specific GlcT1 inhibitor D, L-threo-PDMP at 0.5 mg/ml (*Delgado et al., 2006*) resulted in fewer and larger boutons at NMJ synapses (*Figure 2C,N*), without

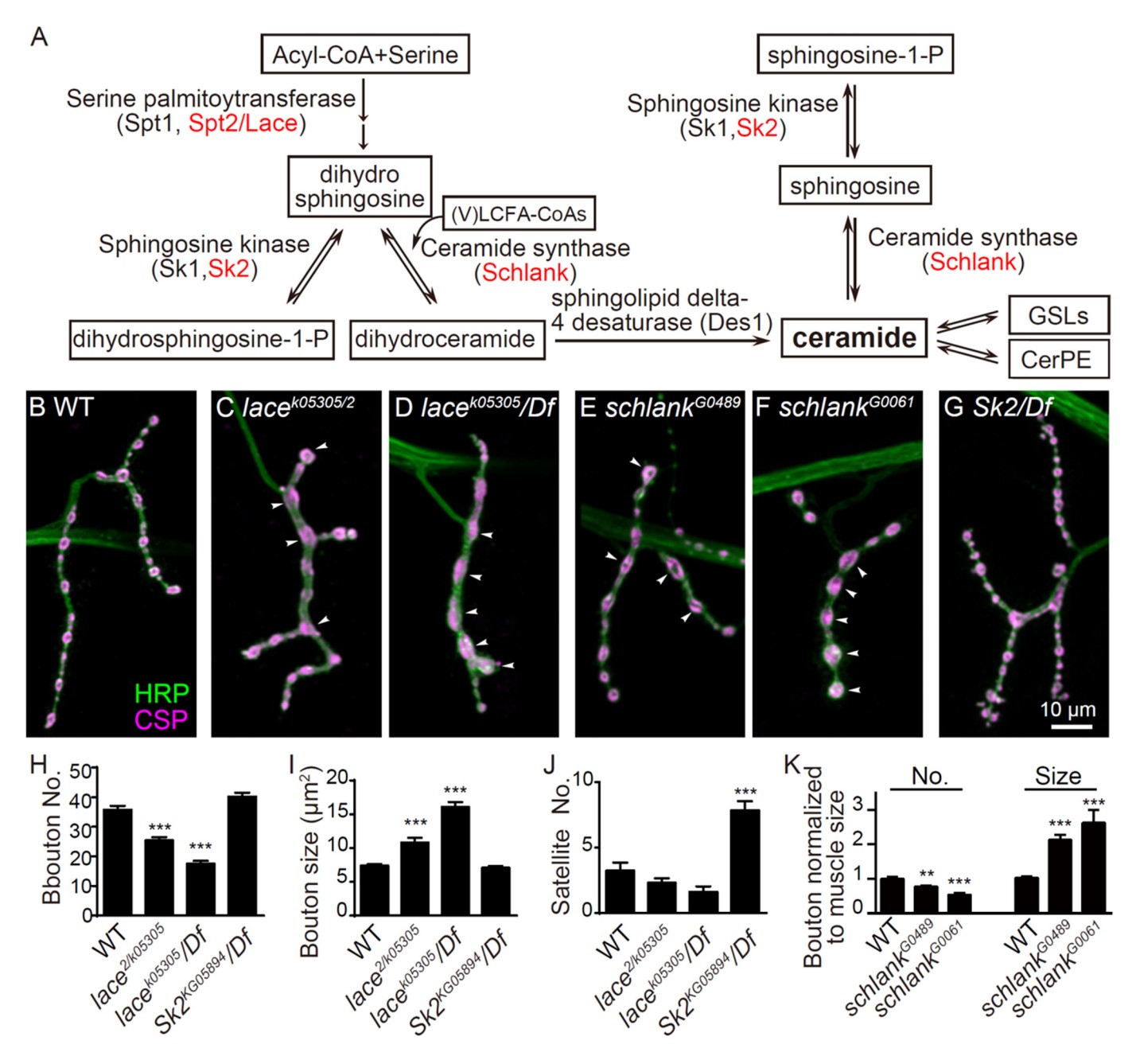

**Figure 1.** NMJ growth depends on de novo synthesis of ceramides (A) Simplified de novo biosynthesis pathway of sphingolipid in *Drosophila* is shown.(*B–G*) Representative images of NMJ4 co-stained with anti-HRP (green) and anti-CSP (magenta) in wild type (B), *lace*$^{k05305}$/*lace*$^2$ (C), *lace*$^{k05305}$/*Df (2L)Exel7063* (D), *schlank*$^{G0489}$/*Y* (E), *schlank*$^{G0061}$/*Y* (F) and *Sk2*$^{KG05894}$/*Df(3L)BSC671* (G). Scale bar: 10 µm; Arrowheads indicate large boutons in different mutants. (*H–J*) Quantifications of bouton number (H), bouton size (I) and satellite bouton number (J) of NMJs in abdominal segments A3 or A4 of different genotypes. (K) Bouton number and size in *schlank* mutants were normalized to muscle surface area as *schlank* mutants showed decreased body size (*Bauer et al., 2009*). *p<0.05; **p<0.01, and ***p<0.001 by student's *t* test between a test genotype and the wild-type control; $n \geq 10$ larvae; error bars: s.e.m. Source data 1. Numerical data for the statistical graphs. The following figure supplement is available for *Figure 1*.

DOI: https://doi.org/10.7554/eLife.38183.002

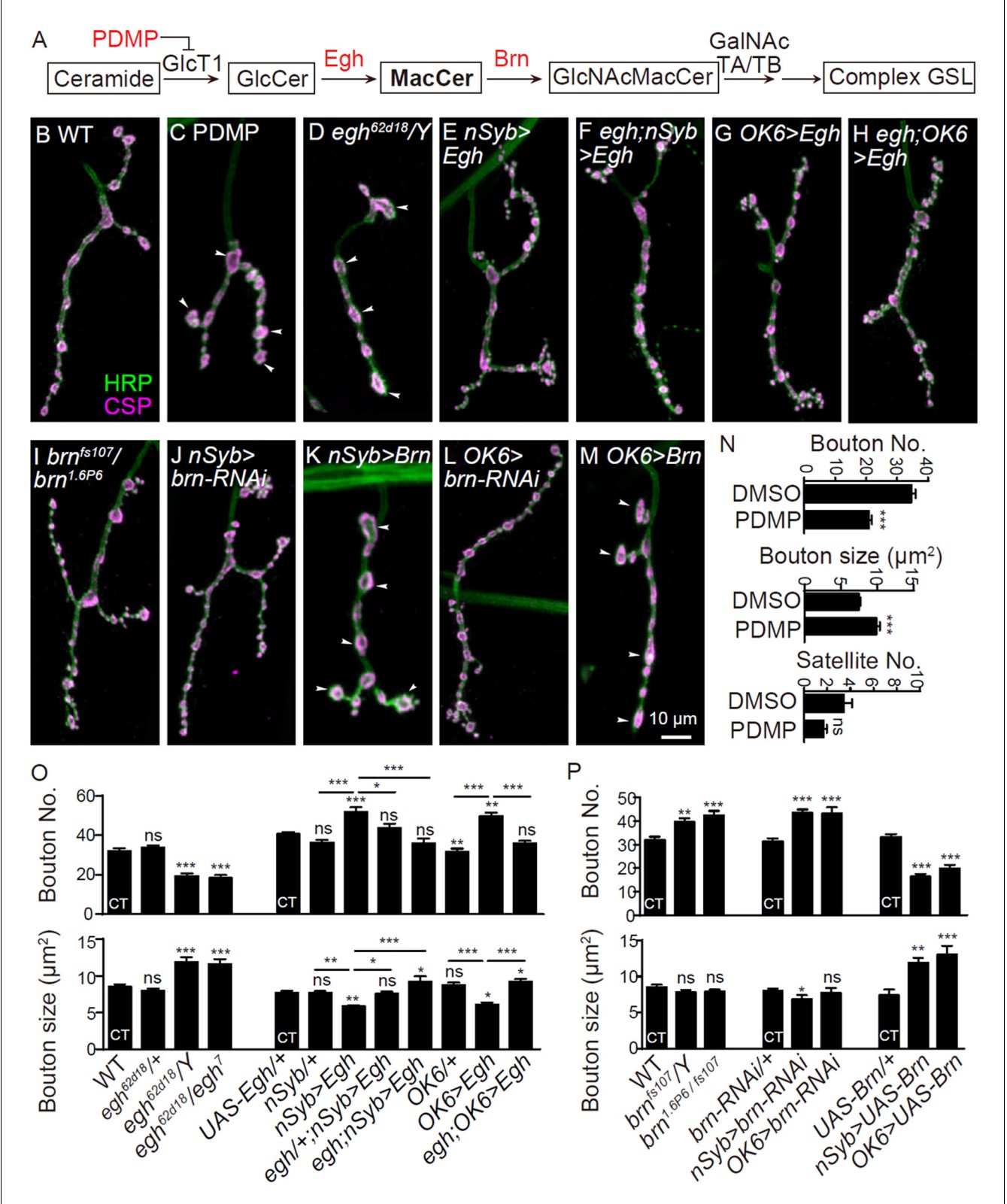

**Figure 2.** GSL synthases Egh and Brn bi-directionally regulates NMJ growth presynaptically  (A) GSL synthesis pathway in *Drosophila*.(*B–M*) Images of NMJ4 co-stained with anti-HRP (green) and anti-CSP (magenta) in wild type (B), larvae treated with 0.5 mg/ml D, L-threo-PDMP (C), *egh^62d18^/Y* (D), *UAS-Egh/+; nSyb-Gal4/+* (E), *egh^62d18^/Y; UAS-Egh/+; nSyb-Gal4/+* (F), *UAS-Egh/OK6-Gal4* (G), *egh^62d18^/Y; UAS-Egh/OK6-Gal4* (H) *brn^fs107^/brn^1.6P6^* (I), *UAS-brn-RNAi/+; nSyb-Gal4/+* (J) *nSyb-Gal4/UAS-brn* (K), *UAS-brn-RNAi/OK6 Gal4* (L), and *OK6-Gal4/+; UAS-brn/+* (M). Scale bar: 10 μm; Arrowheads point

*Figure 2 continued on next page*

*Figure 2 continued*

at large boutons. (**N–P**) Quantifications of total bouton number, bouton size, and satellite bouton number of NMJs in different genotypes or treated with PDMP. 'CT' denotes corresponding control in each multiple comparison.*p<0.05; **p<0.01, ***p<0.001 by one-way ANOVA with Tukey *post hoc* tests; $n \geq 12$ larvae; error bars: s.e.m.

DOI: https://doi.org/10.7554/eLife.38183.003

The following source data and figure supplements are available for figure 2:

**Source data 1.** Numerical data for the statistical graphs.
DOI: https://doi.org/10.7554/eLife.38183.006
**Figure supplement 1.** Additional NMJ images and quantifications.
DOI: https://doi.org/10.7554/eLife.38183.004
**Figure supplement 2.** Alteration of *egh* and *brn* level in glia or muscle does not affect NMJ growth.
DOI: https://doi.org/10.7554/eLife.38183.005

affecting the developmental time and larval size (muscle 4 size was normal; *Figure 2—figure supplement 1*). GlcCer can be further converted into MacCer by Egghead (Egh) (*Wandall et al., 2003*). It has been previously shown that the enzymatic activity of Egh is reduced in *egh*[62d18] and *egh*[7] mutants (*Wandall et al., 2005*). Here we show that both homozygous *egh*[62d18] and hetero-allelic *egh*[62d18]/*egh*[7] mutants also displayed fewer and larger synaptic boutons (*Figure 2,D,O* and *Figure 2—figure supplement 1*; *Huang et al., 2016*). In contrast, neuronal (presynaptic) overexpression of Egh driven by the pan-neuronal *nSyb-Gal4* or motor neuronal specific *OK6-Gal4* led to synaptic overgrowth with more and smaller boutons (*Figure 2, D–H and O* and *Figure 2—figure supplement 1* ), indicating that Egh promotes bouton formation. However, expression of *egh* in muscles (postsynaptic) driven by *C57-Gal4* or in glia driven by *Repo-Gal4* did not affect NMJ growth in both wild type and *egh*[62d18] mutant background (*Figure 2—figure supplement 2*). The cell-type specific manipulations of *egh* expression support that Egh promotes NMJ bouton formation in presynaptic motoneurons. Heterozygous mutation of *egh* significantly suppressed NMJ overgrowth in larvae neuronal overexpressing Egh, and homozygous mutation of *egh* in Egh-overexpressing background fully suppressed bouton number but not bouton size to the control level (*Figure 2O*), indicating that neuronal expression of *egh* promotes NMJ growth in a dose-dependent manner.

Brainiac (Brn) converts MacCer to GlcNAc-MacCer (*Figure 2A*; *Schwientek et al., 2002*). Hypomorphic *brn*[fs.107] and *brn*[1.6P6] mutants display MacCer accumulation in egg chambers (*Pizette et al., 2009*; *Wandall et al., 2005*). We observed more boutons and satellite boutons in homozygous *brn*[fs.107] and trans-allelic *brn*[1.6P6]/*brn*[fs.107] mutants; neuronal knockdown of *brn* by an RNAi under the control of *nSyb-Gal4* or *OK6-Gal4* also resulted in NMJ overgrowth. Conversely, neuronal overexpression of Brn driven by *nSyb-Gal4* or *OK6-Gal4* led to fewer and larger boutons (*Figure 2, I–M and P*; *Huang et al., 2016*), recapitulating the phenotype of *egh* mutants. Furthermore, overexpression and knockdown of *brn* in muscles by *C57-Gal4* or in glia by *Repo-Gal4* did not affect NMJ growth (*Figure 2—figure supplement 2*). The cell type specific rescue and RNAi knockdown results demonstrate that both Egh and Brn act presynaptically. The opposite effect of Egh and Brn in regulating bouton number suggests that the GSL MacCer promotes NMJ growth.

## GSL MacCer promotes NMJ growth

To verify if MacCer indeed promotes NMJ growth, we performed immunostaining and found that endogenous MacCer, detected by a specific anti-MacCer antibody (*Wandall et al., 2003*), was enriched at presynaptic NMJ boutons in a punctate pattern (*Figure 3A*). Compared to wild type, MacCer intensity at NMJs was reduced in *egh* mutant and *brn*-overexpressing larvae; conversely, MacCer intensity was increased at NMJs of *brn* mutant, *brn* RNAi knockdown, and *egh*-overexpressing larvae (*Figure 3, B–H*), demonstrating the specificity of the anti-MacCer. These findings are consistent with previous reports showing that MacCer is decreased in *egh* mutants but increased in *brn* mutants by high performance thin layer chromatography (*Hamel et al., 2010*; *Wandall et al., 2005*), which we confirmed (Figure 8—figure supplement 1). Based on the fact that the presynaptic MacCer levels positively correlate with synaptic bouton number, we conclude that the GSL MacCer promotes NMJ growth presynaptically.

To characterize the synaptic structure in MacCer-deficient NMJs, we examined a few molecules that play important roles in synaptic growth and function, including the cell adhesion molecule

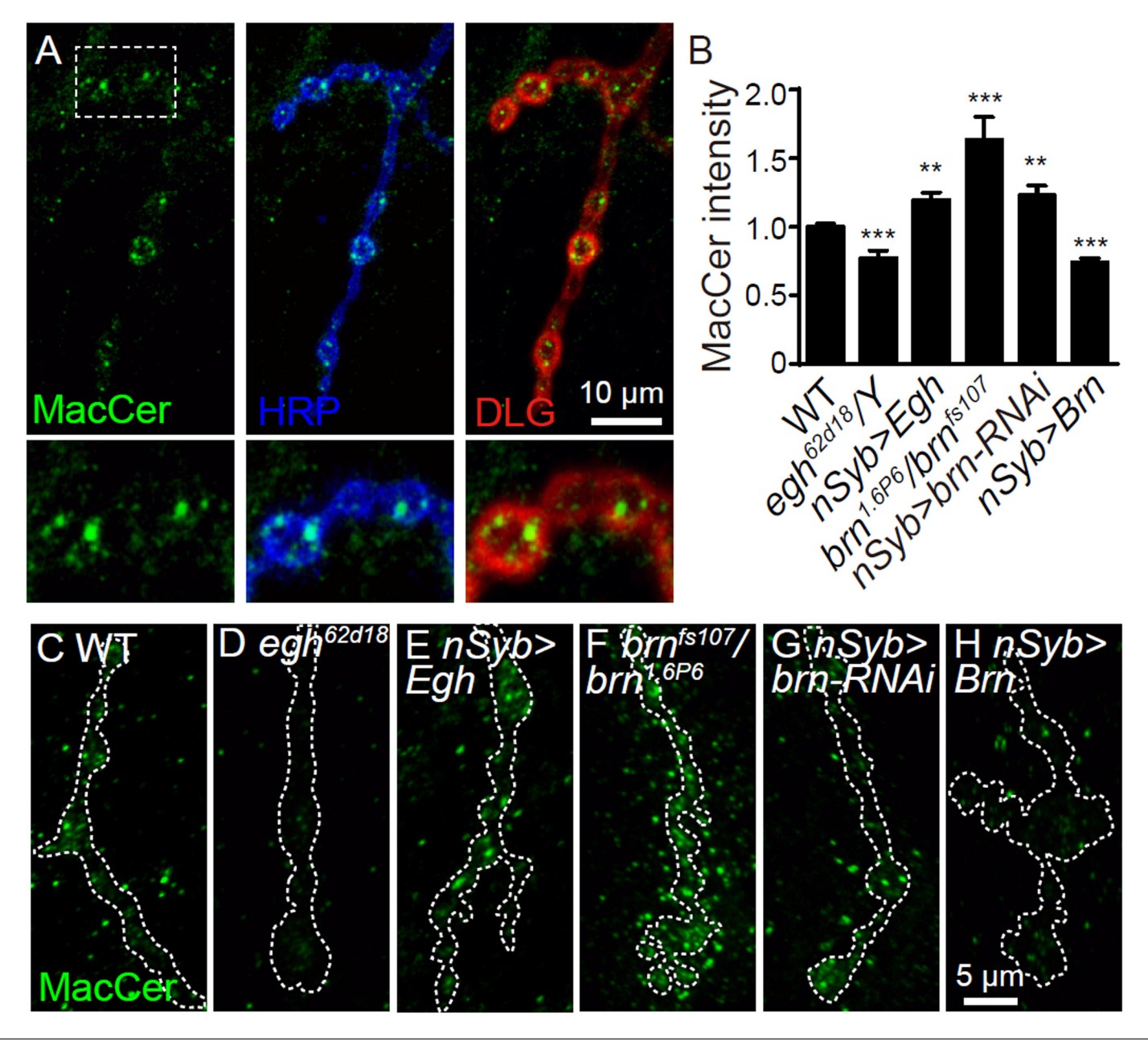

**Figure 3.** MacCer staining intensity at NMJs is bi-directionally regulated by Egh and Brn. (A) Images of wild-type NMJ4 co-stained with anti-MacCer (green), anti-HRP (blue) and anti-DLG (red).MacCer puncta were apparent in presynaptic boutons. Scale bar: 10 μm. (B) Statistical results of normalized intensities of MacCer against anti-HRP staining within presynaptic boutons in different genotypes. **p<0.01 and ***p<0.001 by student's *t* test between a test genotype and wild type; $n \geq 16$ larvae; error bars: s.e.m. (C–H) Representative images of NMJ4 stained with anti-MacCer in wild type (C), *egh⁶²ᵈ¹⁸/Y* (D), *UAS-Egh/+; nSyb-Gal4/+* (E), *brn¹·⁶ᴾ⁶/brnᶠˢ¹⁰⁷* (F), *UAS-brn-RNAi/+; nSyb-Gal4/+* (G) and *nSyb-Gal4/UAS-Brn* (H). Scale bar: 5 μm.

DOI: https://doi.org/10.7554/eLife.38183.007

The following source data and figure supplement are available for figure 3:

**Source data 1.** Numerical data for the statistical graphs.
DOI: https://doi.org/10.7554/eLife.38183.009
**Figure supplement 1.** Images and quantifications of Fas II and Syx1A staining at NMJs of different genotypes.
DOI: https://doi.org/10.7554/eLife.38183.008

Fasciclin II (Fas II) and SNARE protein Syntaxin 1A (Syx1A) (*Schulze et al., 1995*; *Schuster et al., 1996*). The synaptic levels of Fas II and Syx1A were not visibly affected in *egh* mutant and *brn*-over-expressing NMJs (*Figure 3—figure supplement 1*).

## MacCer promotes NMJ bouton formation in a lipid raft-dependent manner

GSLs are constituents of lipid rafts that are detergent-resistant membrane microdomains composed of sphingolipids, sterols and proteins. Like other GSLs, *Drosophila* MacCer is enriched in detergent-insoluble membrane microdomains (*Rietveld et al., 1999*). Consistently, MacCer substantially overlaps with Syx1A, a lipid raft-localized protein in human and *Drosophila* cells (*Chamberlain et al., 2001*; *Zhai et al., 2004*), at NMJs (*Figure 4A*). Our results also show that mutations in *lace* and *schlank* result in fewer boutons at NMJs (*Figure 1*), probably due to a disruption of lipid rafts by depleting total sphingolipids. Thus, we asked if synaptic development requires the formation of lipid rafts. The assembly of lipid rafts is sterol-dependent (*Lingwood and Simons, 2010*); sterol depletion by methyl-β-cyclodextrin (MβCD), a cyclic oligosaccharide that absorbs sterols from the membrane, efficiently disrupts raft assembly in mammalian cells and *Drosophila* cells (*Sharma et al., 2004*; *van Zanten et al., 2009*; *Zhai et al., 2004*). *Drosophila* does not synthesize but takes sterols from food (*Carvalho et al., 2010*). Thus we used drug treatment rather than genetic means to deplete sterols. Feeding wild-type larvae with MβCD at concentrations higher than 25 mM led to early larval lethality, while MβCD treatment at 20 mM led to a mild developmental delay but normal larval and muscle size (*Figure 4—figure supplement 1*). MβCD treatment significantly reduced the co-localization of MacCer and Syx1A (*Figure 4, B and C*). We also treated larvae with another sterol binding drug filipin at 50 μg/ml, which did not affect larval developmental time but led to a mild but significant decrease in muscle size (*Figure 4—figure supplement 1*). Treatments with both drugs resulted in obviously fewer and larger boutons at NMJs, recapitulating the NMJ phenotype of sphingolipid- and MacCer-deficient larvae (*Figure 4, D–G* and *Figure 4—figure supplement 1*). These results suggest that a proper level of sterols is required for bouton formation.

Depleting sterol with MβCD in *egh*-overexpressing background fully restrict the NMJ overgrowth to the level of the wild-type larvae treated with MβCD. Moreover, MβCD treatments in sphingolipid-deficient *lace*^*k05305*/*Df* mutants and MacCer-deficient larvae (*egh*^*62d18* and *nSyb-Gal4*/*UAS-brn*) did not exacerbate the NMJ phenotype (*Figure 4G*, and *Figure 4—figure supplement 1*), suggesting that sterol may modulate NMJ growth in a common genetic pathway with sphingolipids and MacCer. We then determined if MβCD treatment suppresses NMJ growth via directly reducing the MacCer level. We found that MβCD completely suppressed the NMJ overgrowth without detectably reducing the synaptic MacCer level in wild-type and Egh-overexpressing larvae; similarly, Filipin treatment did not affect the synaptic level of MacCer (*Figure 4, H–N*).These data suggest that restriction of synaptic growth by MβCD may be due to defects of MacCer function by disrupting raft assembly rather than a reduction in the synaptic level of MacCer per se. This finding is consistent with a previous report that disrupting the formation of membrane rafts may compromise the function of raft-associated factors but not necessarily affecting their levels; more specifically, the localization of a raft protein LFA-1 was unaltered examined by confocal microscopy at subcellular level but altered by single-molecule near-field optical microscopy at molecular level upon raft disruption (*van Zanten et al., 2009*). Together, these data suggest that the NMJ growth-promoting effect of MacCer depends on normal lipid rafts.

## MacCer promotes NMJ bouton formation via Wg signaling

NMJ development is controlled in part by growth factors and pathways (*Bayat et al., 2011*; *Harris and Littleton, 2015*; *Korkut and Budnik, 2009*). Bone morphogenic protein (BMP) is an important growth promoting signaling in NMJs (*Bayat et al., 2011*). However, we found that the NMJ growth-promoting effect of MacCer is not mediated through the BMP signaling pathway (*Huang et al., 2016*). Wnt1/Wingless (Wg) activates another signaling pathway promoting NMJ growth (*Korkut and Budnik, 2009*). Because Wg is a lipid raft-associated protein (*Zhai et al., 2004*), we further investigated if Wg signaling play roles in MacCer-mediated NMJ growth. Previous reports showed that loss-of-function mutations of Wg signaling components, including the ligand Wg, the receptor dFrizzled2 (Fz2), the co-receptor Arrow (Arr), and the Wg-binding protein Evenness

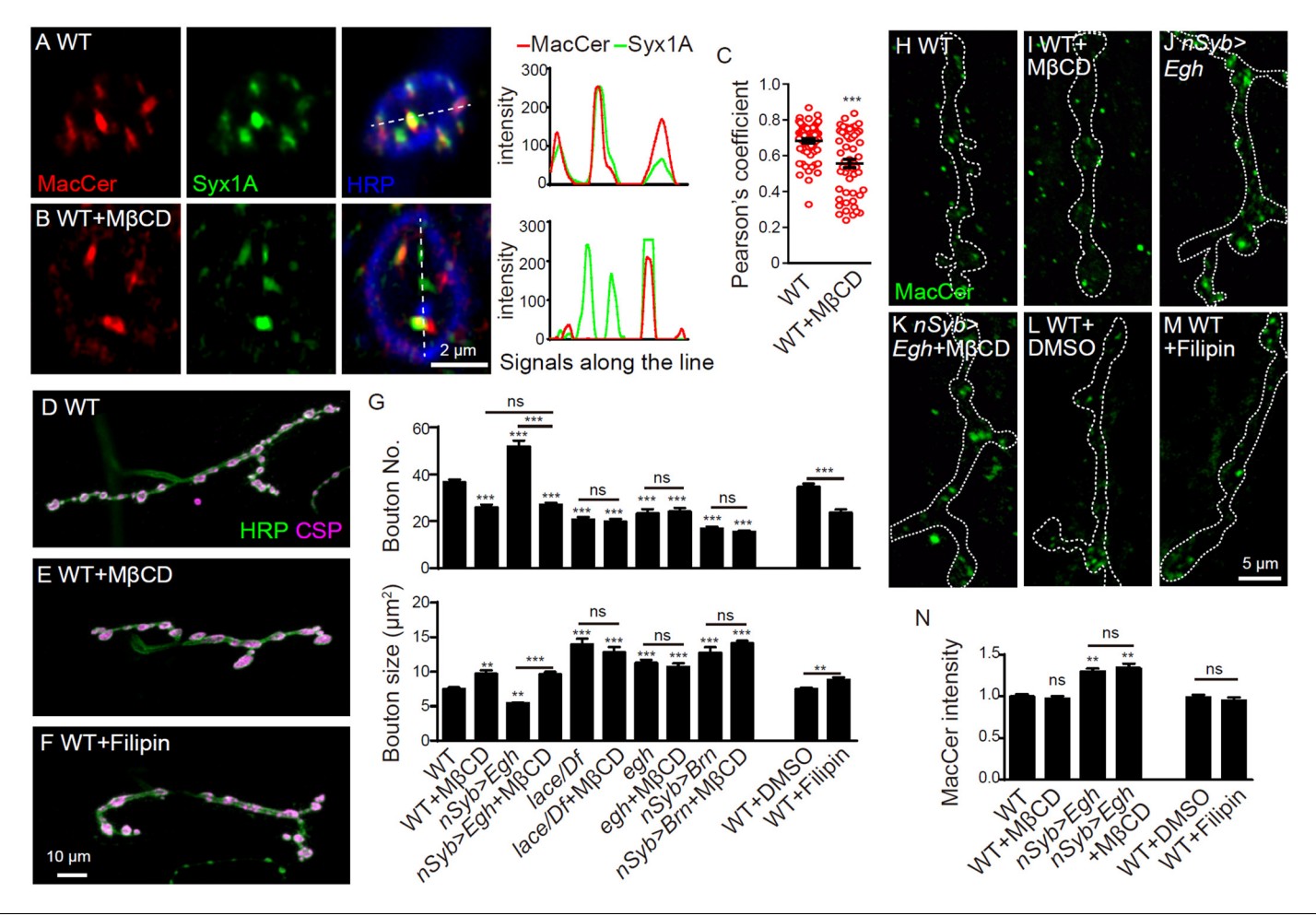

**Figure 4.** Sterol-depletion inhibits NMJ growth in a common genetic pathway with MacCer. (**A and B**) Confocal images of single slice of NMJ4 boutons triple-labeled with anti-MacCer (red), anti-Syx1A (green) and anti-HRP (blue) in wild type with or without 20 mM MβCD treatment. Plot profiles of the relative intensity along the dashed lines are shown. (**C**) Pearson's coefficients of colocalization between MacCer and Syx1A. n = 63 and 52 boutons from eight wild-type larvae each with or without MβCD treatment. (**D–F**) Images of NMJ4 co-labeled for anti-HRP (green) and anti-CSP (magenta) in untreated wild type (**D**), wild type treated with 20 mM MβCD (**E**), wild type treated with 50 μg/ml filipin III (**F**), Scale bar: 10 μm. (**G**) Quantification of bouton number and bouton size of NMJs. ns, no significance, ***p<0.001 by one-way ANOVA with Tukey *post hoc* tests, error bars: s.e.m. (**H–M**) Images of NMJs from larvae stained with anti-MacCer of wild type (**H**), wild type treated with 20 mM MβCD (**I**), *UAS-Egh/+; nSyb-Gal4/+* (**J**), and *UAS-Egh/+; nSyb-Gal4/+* treated with 20 mM MβCD (**K**), wild-type larvae treated with vehicle DMSO (**L**) or 50 μg/ml filipin III (**M**). Scale bar: 5 μm. (**N**) Quantification of MacCer intensities normalized to HRP intensities in different genotypes. ns, no significance; **p<0.01 by one-way ANOVA with Tukey *post hoc* tests; n ≥ 12 larvae; error bars: s.e.m.

DOI: https://doi.org/10.7554/eLife.38183.010

The following source data and figure supplement are available for figure 4:

**Source data 1.** Numerical data for the statistical graphs.
DOI: https://doi.org/10.7554/eLife.38183.012

**Figure supplement 1.** Additional NMJ images and quantifications of bouton number and bouton size.
DOI: https://doi.org/10.7554/eLife.38183.011

Interrupted/Wntless (Evi/Wls), decrease synaptic bouton number, while overexpression of Wg in motor neurons results in synaptic overgrowth (*Korkut et al., 2009*; *Miech et al., 2008*; *Packard et al., 2002*). In addition, the proper secretion, binding with the receptor Frizzled, and activation of Wg requires lipidation at the conserved serine at amino acid 239 (*Janda et al., 2012*; *Takada et al., 2006*). Specifically, the synaptic level of Wg and the downstream signaling are

positively regulated by the acyltransferase Porcupine (*Packard et al., 2002*), which activates the lipidation and raft-association of Wg (*Zhai et al., 2004*). We hypothesized that the NMJ growth-promoting effect of Wg depends on its association with membrane rafts. Indeed, raft disruption by MβCD treatment which depletes sterol significantly reduced Wg staining intensity at NMJs and constrained NMJ overgrowth in Wg-overexpressing larvae (*Figure 5—figure supplement 1*), suggesting that normal Wg signaling at NMJs requires intact lipid rafts.

Next, we determined whether Wg signaling cooperates with MacCer in promoting NMJ growth. Genetic analysis showed that loss of one copy of $egh^{62d18}$ had no effect on NMJ morphology but fully suppressed the NMJ overgrowth phenotype of Wg-overexpressing larvae. Homozygous mutation of *egh* or overexpression of *brn* on a Wg-overexpressing background led to a decrease in bouton number, recapitulating the phenotype of *egh* mutants and *brn*-overexpressing larvae (*Figure 5, A–E, K* and *Figure 5—figure supplement 1*). These results together indicate that the NMJ growth-promoting effect of Wg signaling depends on the proper level of MacCer. Conversely, mutation in $wg^1/wg^{CX4}$ restricted NMJ overgrowth in Egh-overexpressing larvae to the *wg* mutant level (*Figure 5, G–I and K*). In addition, although bouton number was normal in $wg^1$ and $egh^{62d18}$ single heterozygotes ($wg^1$/+and $egh^{62d18}$/+), there were fewer and larger boutons in $egh^{62d18}$/+; $wg^1$/+ transheterozygotes. Similarly reduced bouton numbers were observed in $egh^{62d18}$/+; $arr^{k08131}$/+and $egh^{62d18}$/+; $evi^2$/+ transheterozygotes (*Figure 5, J, K* and *Figure 5—figure supplement 1*), suggesting that Egh and Wg signaling act in the same genetic pathway. Together, these results demonstrate that MacCer works synergistically with Wg signaling in promoting NMJ growth and bouton formation.

## MacCer facilitates local presynaptic Wg signaling at NMJs

To reveal the molecular mechanism by which MacCer promotes NMJ growth via Wg signaling, we first examined if there was colocalization between MacCer and Wg. Indeed, a substantial overlap between MacCer and Wg immunoreactivity was observed in NMJ boutons (*Figure 6A and E*). It is known that Wg secretion from presynaptic terminals requires the recycling endosomal small GTPase Rab11 (*Koles et al., 2012*; *Korkut et al., 2009*). It is also known that recycling endosomes are enriched with components of lipid rafts including Lactosylceramide (LacCer), the vertebrate analog of MacCer (*Balasubramanian et al., 2007*; *Gagescu et al., 2000*; *Hortsch et al., 2010*). Consistently, we found strong colocalization of MacCer with Rab11 at NMJ terminals (*Figure 6, C and E*). In contrast, MacCer puncta did not overlap with the early endosomal marker Rab5-YFP or the late endosome protein Spinster-GFP (Spin-GFP). Although GSLs are synthesized in the Golgi apparatus, there were few overlaps between MacCer and the Golgi marker Mannose II-GFP (*Figure 5—figure supplement 1*).

The specific colocalization of MacCer with Wg and Rab11 suggests that the trafficking or distribution of Wg at NMJ synapses may be altered upon MacCer reduction. Indeed, the staining intensity of endogenous Wg was significantly decreased in $egh^{62d18}$ mutant boutons and *brn*-overexpressing NMJs; Conversely, Wg intensity was increased in *egh*-overexpressing and *brn*-RNAi knockdown boutons (*Figure 6B, F–I* and *Figure 6—figure supplement 2*). Thus, MacCer is both required and sufficient for Wg level at NMJ synapses. Though the total protein level of Wg in *egh* mutant brains appeared normal by Western assay, we found an increase of Wg in the central neuropil but not in peripheral axons within segmental nerves of *egh* mutants (*Figure 6—figure supplement 3*), suggesting a defect in axonal transport of Wg. In contrast to reduced synaptic Wg, the staining intensities of Rab11 and the Wg receptor Fz2 were largely normal in *egh* NMJs (*Figure 6D* and *Figure 6—figure supplement 4*). Conversely, MacCer staining was normal in *wg* or *rab11* mutants (*Figure 6—figure supplement 4*). These results together show that MacCer positively regulates Wg level, but not vice versa, at NMJs.

Wg coordinates pre- and postsynaptic signaling cascades that modulate synapse development. The presynaptic Wg signaling pathway promotes formation of Futsch-associated microtubule loops without transcriptional regulation (*Korkut and Budnik, 2009*; *Miech et al., 2008*). Futsch is a microtubule-associated protein, and the formation of Futsch-positive microtubule loops is essential for synaptic growth (*Roos et al., 2000*). In *wg* and *arr* mutants with impaired Wg signaling, the number of Futsch loops is reduced while the percentage of boutons containing unbundled Futsch is increased (*Miech et al., 2008*; *Packard et al., 2002*). We observed similar Futsch staining phenotype in *brn*-overexpression larvae, $egh^{62d18}$ mutants and $egh^{62d18}$/+; $wg^1$/+ transheterozygotes, i.e., a

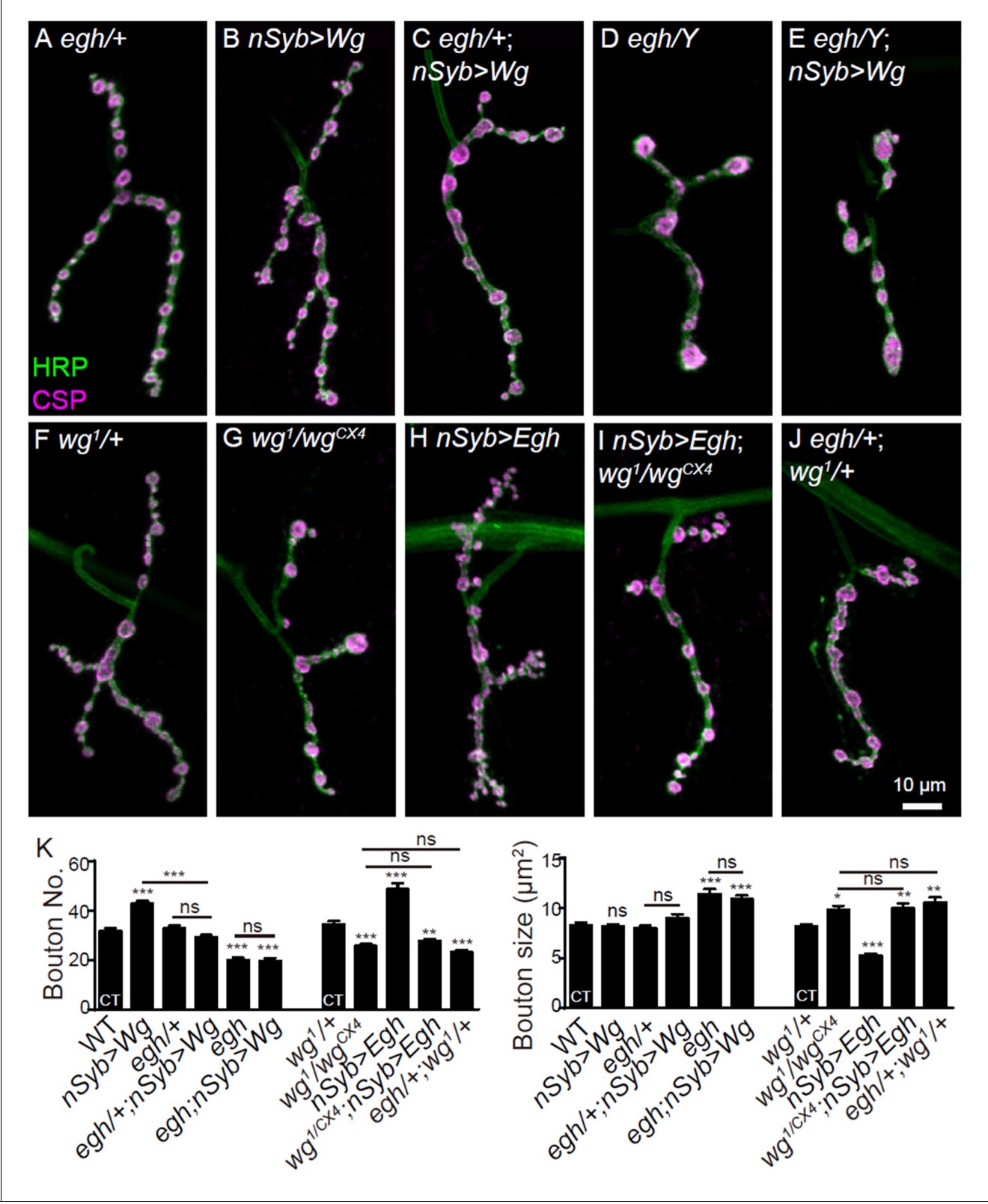

**Figure 5.** MacCer is required for the NMJ growth-promoting effect of Wg signaling. (A–J) Representative images of NMJ4 co-stained with anti-HRP (green) and anti-CSP (magenta) in *egh^62d18/+* (control, (A), *UAS-Wg-HA/+; nSyb-Gal4/+* (B), *egh^62d18/+; UAS-Wg-HA/+; nSyb-Gal4/+* (C), *egh^62d18* (D), *egh^62d18; UAS-Wg-HA/+; nSyb-Gal4/+* (E), *wg^1/+* (F), *wg^1/wg^CX4* (G), *nSyb-Gal4/UAS-Egh* (H), *wg^CX4,UAS-Egh/wg^1; nSyb-Gal4/+* (I), and *egh^62d18/+; wg^1/+* (J).Scale bar: 10 µm. (K) Quantifications of bouton number and bouton size of NMJs in different genotypes. 'CT' denotes corresponding control

*Figure 5 continued on next page*

*Figure 5 continued*

in each multiple comparison. $n \geq 12$ larvae; ns, no significance, *p<0.05; **p<0.01; ***p<0.001 by one-way ANOVA with Tukey *post hoc* tests, error bars: s.e.m.

DOI: https://doi.org/10.7554/eLife.38183.013

The following source data and figure supplements are available for figure 5:

**Source data 1.** Numerical data for the statistical graphs.
DOI: https://doi.org/10.7554/eLife.38183.016
**Figure supplement 1.** Additional NMJ images and quantifications.
DOI: https://doi.org/10.7554/eLife.38183.014
**Figure supplement 2.** Additional NMJ images and quantifications.
DOI: https://doi.org/10.7554/eLife.38183.015

significant increase in the percentage of boutons containing unbundled and punctate Futsch signals, especially in large boutons (*Figure 6, J–M* and *Figure 6—figure supplement 2*), indicating that MacCer and Wg cooperate in controlling the formation of Futsch loops. Together, these results support that MacCer facilitates presynaptic Wg signaling during NMJ development.

## MacCer is not required for postsynaptic differentiation

Distinct from the presynaptic Wg pathway, Wg regulates postsynaptic differentiation and bouton formation by activating the Fz2 nuclear import pathway in postsynaptic muscles (*Ataman et al., 2006*; *Mathew et al., 2005*; *Mosca and Schwarz, 2010*); defects in this pathway lead to a reduced number of boutons, altered distribution of the postsynaptic type A glutamate receptor (GluRIIA), a decrease of subsynaptic reticulum (SSR) with increase of 'ghost bouton' that lacks of Discs large (DLG) postsynaptic staining, and enlarged pockets in SSR juxtaposed to active zones (*Mosca and Schwarz, 2010*; *Packard et al., 2002*). However, we did not find these abnormalities in $egh^{62d18}$ mutants except that the GluRIIA intensity was significantly but slightly increased (*Figure 7, A, B and E*); rather, the ghost bouton number, the GluRIIA cluster size, as well as the SSR ultrastructure were largely normal (*Figure 7, C–I*), suggesting that the postsynaptic Wg signaling remains unaffected upon MacCer reduction.

## Wg contains a functional MacCer-binding domain

The results shown above suggest that MacCer may interact with Wg and thus modulate its signaling activity. Previous in vitro studies show that GSLs are able to specifically bind proteins containing a structurally conserved GSL-binding motif (GBM), which contains both basic (Lys, Arg) and turn-inducing (Gly, Ser) residues (*Fantini and Yahi, 2015*; *Hamel et al., 2010*). By in silico analysis, we identified a potential GBM of Wg between amino acid residues 233 and 247, therein Lys and Met are required for the interaction with MacCer (*Figure 8A*). Interestingly, this region contains a palmitoylation site at Ser-239 by a palmitoleate (C16:1; *Figure 8A and C*) (*Herr and Basler, 2012*; *Kakugawa et al., 2015*; *Takada et al., 2006*), which is known to affect the association of membrane proteins to lipid rafts (*Resh, 2004*). Structure homology modeling and a series of molecular dynamics simulations showed that this acylated GBM is fully exposed on the protein surface and perfectly fit with MacCer. A remarkable feature of the molecular complex is that the ceramide part of MacCer interacts with the acyl chain of the GBM, whereas the sugars of MacCer bind the peptidic part of the GBM (*Figure 8B and C*).

We then analyzed the interaction of a synthetic peptide derived from the predicted GBM of Wg with monolayers of purified GSLs by Langmuir microtensiometer experiments (*Di Scala et al., 2014*). The amino acid sequence of the peptide contains four Cys residues with several possibilities of forming two disulfide bridges may not fully stable under our experimental conditions. Thus we designed a synthetic peptide in which the Cys residues were replaced by isosteric Ser residues (referred to as $Wg^{233-247}$). This Cys/Ser substitution respects the electrostatic distribution of partial charges at the surface of the peptide. The $Wg^{233-247}$ peptide showed high affinity with LacCer (*Figure 8—figure supplement 1*).

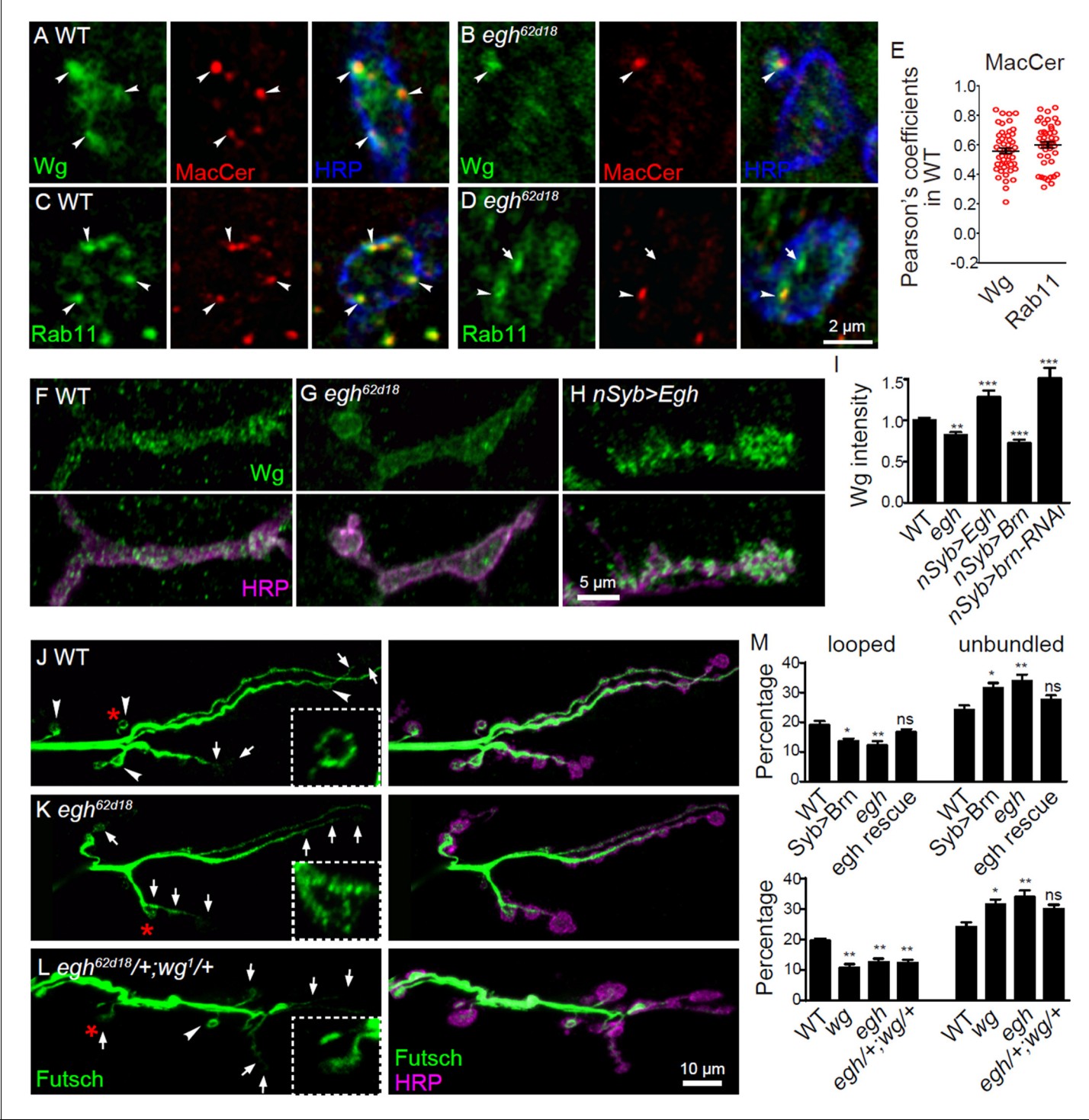

**Figure 6.** MacCer is required for the local presynaptic Wg signaling at NMJs. (A–D) Confocal images of single slice of NMJ4 boutons triple-labeled with anti-MacCer (red), anti-HRP (blue) and anti-Wg (green; A–B) or anti-Rab11 (green; C–D) in wild type and $egh^{62d18}$ mutants. MacCer puncta showed substantial colocalization (arrowheads) with Wg and Rab11 in boutons. Arrows indicate Rab11-positive puncta without MacCer staining in the $egh^{62d18}$ mutants. Images were processed by deconvolution. Scale bar: 2 µm. (E) Pearson's coefficients of colocalization between MacCer and indicated proteins. n = 51 and 44 boutons (from ten wild-type larvae each) for colocalization of MacCer with Wg and Rab11, respectively. (F–H) Representative images of NMJ4 co-labeled with anti-Wg (green) and anti-HRP (magenta) in wild type (F), $egh^{62d18}$ mutants (G) and $UAS$-$Egh/+$; $nSyb$-$Gal4/+$(H). Scale bar: 5 µm. (I) Quantification of intensities of endogenous Wg normalized to HRP intensities in different genotypes. $n \geq 15$ larvae; ns, no significance; **p<0.01 and ***p<0.001 by student's $t$ test; error bars: s.e.m. (J–L) Representative images of NMJ6/7 labeled with anti-Futsch (green) and anti-HRP

*Figure 6 continued on next page*

*Figure 6 continued*

(magenta) in wild type (J), *egh*[62d18] mutants (K) and *egh*[62d18]/+; *wg*[1]/+transheterozygotes (L). Insets show higher magnification images of Futsch staining of a single bouton (asterisk). Arrows indicate boutons displaying unbundled Futsch; arrowheads denote boutons with Futsch loops. Scale bar: 10 μm. (M) Quantifications of percentage of Futsch loops and unbundled Futsch staining in different genotypes. The genotype of *egh* rescue was *egh*[62d18]; *UAS-Egh*/+; *nSyb-Gal4*/+; the genotype of *wg* was *wg*[1]/*wg*[CX4]. $n \geq 12$ larvae; ns, no significance, *p<0.05; **p<0.01 by one-way ANOVA with Tukey *post hoc* tests, error bars: s.e.m.

DOI: https://doi.org/10.7554/eLife.38183.017

The following source data and figure supplements are available for figure 6:

**Source data 1.** Numerical data for the statistical graphs.
DOI: https://doi.org/10.7554/eLife.38183.022
**Figure supplement 1.** MacCer puncta showed few overlap with Rab5-YFP, Spinster-GFP and Mannose II-GFP.
DOI: https://doi.org/10.7554/eLife.38183.018
**Figure supplement 2.** The Wg level in *egh* brains by immunochemical analysis.
DOI: https://doi.org/10.7554/eLife.38183.019
**Figure supplement 3.** Additional images showing Wg and Futsch staining at NMJ of different genotypes.
DOI: https://doi.org/10.7554/eLife.38183.020
**Figure supplement 4.** Images and quantifications of MacCer, Rab11, and Fz2 staining at NMJ of different genotypes.
DOI: https://doi.org/10.7554/eLife.38183.021

We further analyzed the interaction of Wg[233-247] with GSLs extracted from *Drosophila* larvae of different genotypes. In agreement with previous findings (*Hamel et al., 2010*; *Wandall et al., 2005*), wild-type larvae expressed a broad range of GSLs including tetra- and penta-hexosylceramides, while *egh*[62d18] mutants lacked MacCer and *brn*[1.6P6] mutants contained almost exclusively MacCer (*Figure 8—figure supplement 1*). As expected, Wg[233-247] showed strong interaction with MacCer-enriched GSLs from *brn* mutants but no interaction with MacCer-deficient GSLs from *egh* mutants (*Figure 8D*). In contrast, a GBM mutant containing mutations at key Lys and Met (K234E and M238R; *Figure 8A*) showed no interaction to GSLs from *brn* mutants (*Figure 8—figure supplement 1*). Moreover, we examined whether full-length Wg binds to MacCer-enriched GSLs. Myc-tagged full-length Wg was produced by an in vitro translation kit containing rabbit reticulocyte lysates and purified by anti-Myc immunoprecipitation. Wild-typed Wg showed high affinity to GSLs from *brn* but not *egh* mutants; in contrast, Wg with GBM deleted or mutated showed no affinity to GSLs from *brn* mutants (*Figure 8E and F*).

We further validated the GSL binding ability of Wg by pull-down assay. Wild-type Myc-Wg but not Wg with mutated GBM was substantially pulled down by LacCer-coated beads (*Figure 8G*). Taken together, these data support that Wg has an intrinsic affinity for MacCer/LacCer via GBM.

## Wg GBM is critical for its colocalization with MacCer and normal NMJ growth

To further test the physiological role of the Wg GBM, we generated double site-directed point mutations of K234E and M238R in Wg (*Figure 8A*) by CRISPR/Cas9-mediated mutagenesis. This double point-mutation *wg* allele was named *wg*[GBM]. Homozygous *wg*[GBM] or in combination with a null allele *wg*[CX4] (*wg*[GBM]/*wg*[CX4]) led to larval lethality before 3[rd] instar stage. However, *wg*[GBM] in combination with a hypomorphic allele *wg*[1] (*wg*[1]/*wg*[GBM]) resulted in escapers survived to adults. The Wg protein level in *wg*[1]/*wg*[GBM] mutants was largely normal in whole larvae by western analysis and at NMJs by immunostaining compared with *wg*[1] heterozygotes (*wg*[1]/+) (*Figure 9, A–D*). MacCer level was not altered in *wg*[1]/*wg*[GBM] mutants compared with that of *wg*[1]/+ heterozygous control (*Figure 9, A–C*). However, the colocalization between Wg and MacCer was significantly decreased in *wg*[1]/*wg*[GBM] mutants compared with that of *wg*[1]/+ heterozygous and wild-type control (*Figure 9, E–H*), suggesting that *wg*[GBM] mutation, presumably a null allele, affect proper localization of Wg in synaptic boutons.

To determine if Wg GBM interacts with MacCer in regulating Wg signaling and bouton formation, we examined the genetic interaction between *wg*[GBM] and *egh* mutation. We found that *wg*[GBM]/*wg*[1] mutants and *egh*[62d18]/+; *wg*[GBM]/+ transheterozygotes showed fewer and larger boutons, and the

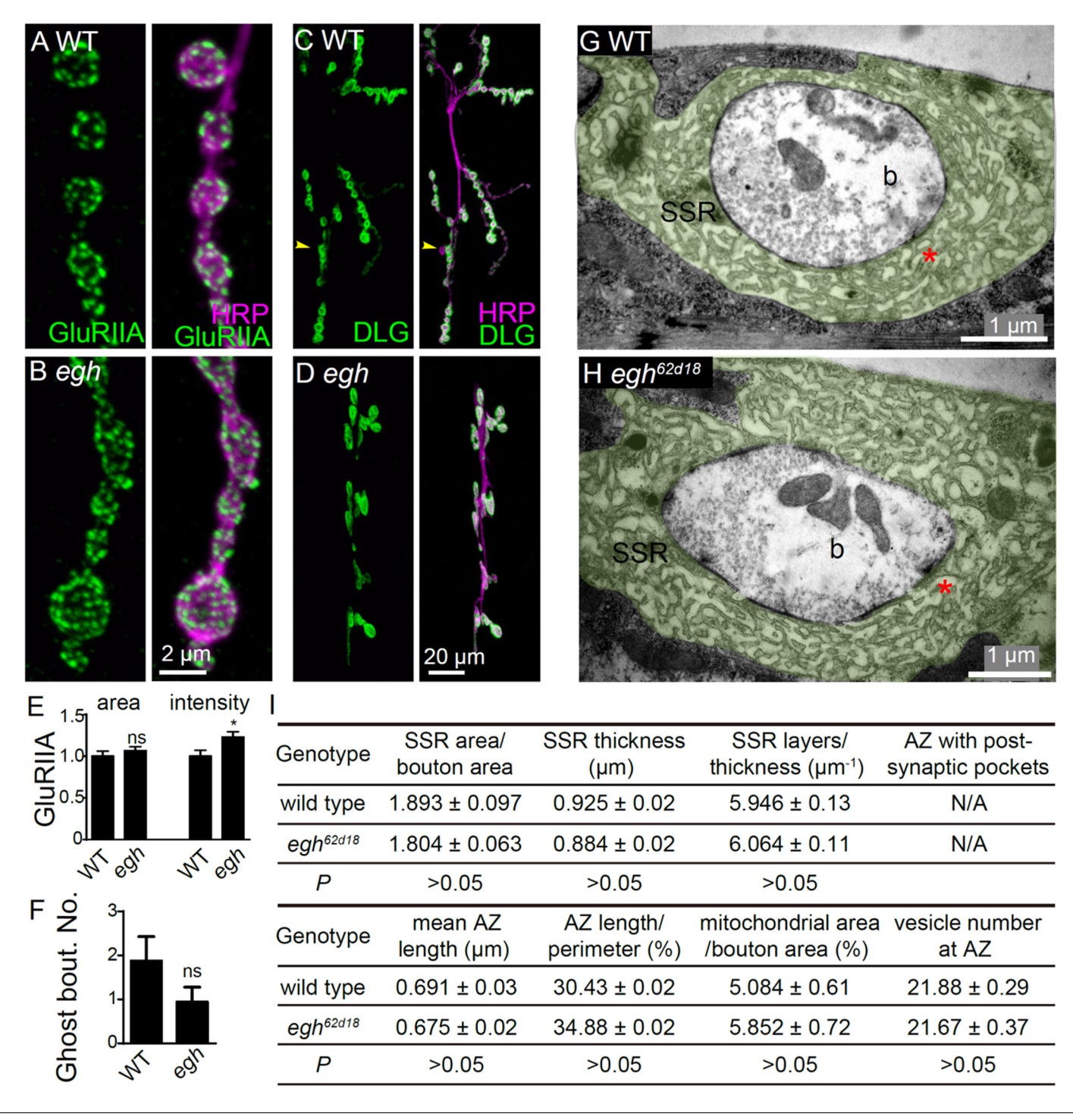

**Figure 7.** Postsynaptic differentiation is normal in *egh* mutants. (**A and B**) Representative images of NMJ4 from wild type and *egh*[62d18] mutants co-stained with anti-HRP (magenta) and anti-GluRIIA (green). Scale bar: 2 μm. (**C and D**) Images of NMJ6/7 from wild type and *egh*[62d18] mutants co-stained with anti-HRP (magenta) and anti-DLG (green). An arrowhead points at a ghost bouton. Scale bar: 20 μm. (**E and F**) Quantification of normalized intensities and area of GluRIIA against anti-HRP staining of NMJ4 (**E**) and ghost bouton number of NMJ6/7 (**F**) in wild type and *egh*[62d18] mutants. ns, no significance, *p<0.05 by student's *t* test; *n* ≥ 12 larvae; error bars: s.e.m. (**G and H**) Ultrastructure images of NMJ6/7 boutons from wild type and *egh*[62d18] mutants. The subsynaptic reticulum (SSR; green) is folded membrane network that surrounds the presynaptic bouton (b); The SSR region juxtaposed to the active zone (AZ) is indicated by asterisks in red. Scale bar: 1 μm. (**I**) Quantification of various bouton parameters of NMJ 6/7 synapses

*Figure 7 continued on next page*

*Figure 7 continued*
in wild type and *egh*[62d18] mutants. n = 50 boutons from five wild-type animals, and n = 37 boutons **from three** egh[62d18] larvae. Statistical analysis was performed by student's *t* test.
DOI: https://doi.org/10.7554/eLife.38183.023
The following source data is available for figure 7:
**Source data 1.** Numerical data for the statistical graphs.
DOI: https://doi.org/10.7554/eLife.38183.024

phenotype in *wg*[GBM]/*wg*[1] mutants was rescued by neuronal expression of Wg but not by expression of Egh (*Figure 9I–N*). The NMJ phenotype of *wg*[GBM]/*wg*[1] was insensitive to *egh* overexpression (*Figure 9L and N*), supporting that *wg* is epistatic to *egh*. We also observed significantly fewer boutons with Futsch loops and more boutons with unbundled Futsch staining in both *wg*[GBM]/*wg*[1] mutants and *egh*[62d18]/+; *wg*[GBM]/+ transheterozygotes (*Figure 9O-R*). Together, these findings suggest that the intact Wg GBM is critical for proper Wg localization, Wg signaling and thus bouton formation.

## Discussion

In the present study, based on genetic analysis of cell-type specific manipulations of Egh, Brn and Wg expression, together with immunochemical, genetic interaction and lipid-protein interaction analysis, we demonstrate that MacCer promotes NMJ bouton formation by interacting with Wg, thereby facilitating presynaptic Wg signaling. This is the first report defining a critical role of a specific class of sphingolipids in synapse development.

### A crucial role for sphingolipids in NMJ synapse development

Sphingolipids are essential components of lipid rafts which regulate multiple signaling pathways and neural functions. The present study provides compelling evidence implicating sphingolipids in regulating NMJ growth (*Figure 1* and *Figure 2*; *Supplementary file 1*). Previous studies revealed that various mutations in either *egh* or *brn* result in similar ovarian defects and neuronal hypertrophy during embryogenesis (*Goode et al., 1996*). The similarity of *egh* and *brn* mutant phenotypes suggests that both are caused by a lack of complex GSLs downstream of *brn*. A more recent study showed that *egh* mutations result in overgrowth of subperineurial glia through elevated phosphatidylinositol-3 kinase (PI3K) signaling; the glial phenotype is due to the accumulation of GlcCer, the substrate of Egh (*Dahlgaard et al., 2012*). At NMJs, however, our data demonstrate that the NMJ defects in *egh* mutants are caused by lack of MacCer.

Do other classes of sphingolipids besides MacCer also regulate synapse development? Many sphingolipids, such as phosphorylated sphingosines and CerPE, the most abundant sphingolipid in lipid rafts, play a diverse array of functions in the nervous system (*Fantini and Yahi, 2015*; *Yonamine et al., 2011*). It will be intriguing to define the molecular mechanisms by which these structural and signaling sphingolipids regulate synapse development.

### MacCer promotes synaptic growth via presynaptic Wg signaling

*Drosophila* NMJ development is mediated by multiple signaling cascades. How does MacCer function in synaptic growth? Consistent with previous findings that GSL/lipid rafts are enriched in recycling endosomes (*Balasubramanian et al., 2007*; *Gagescu et al., 2000*; *Hortsch et al., 2010*), we observed a substantial colocalization between MacCer and recycling endosomal marker Rab11 at NMJ synapses (*Figure 6*). We note, however, the impact of MacCer and Rab11 on NMJ growth is opposite. The NMJ bouton formation is positively regulated by MacCer but negatively regulated by Rab11 (*Huang et al., 2016*; *Khodosh et al., 2006*; *Liu et al., 2014*; this study). The opposite effect on NMJ growth of the two might be caused by the fact that MacCer and Rab11 regulate different signaling pathways. For example, Rab11 inhibits BMP signaling (*Huang et al., 2016*; *Liu et al., 2014*) while MacCer facilitates Wg signaling (this study) at NMJ terminals.

Our results showed that MacCer facilitates the presynaptic Wg signaling in promoting NMJ growth (*Figure 5*). In contrast to the abnormal microtubule cytoskeleton (*Figure 6*), *egh* mutants

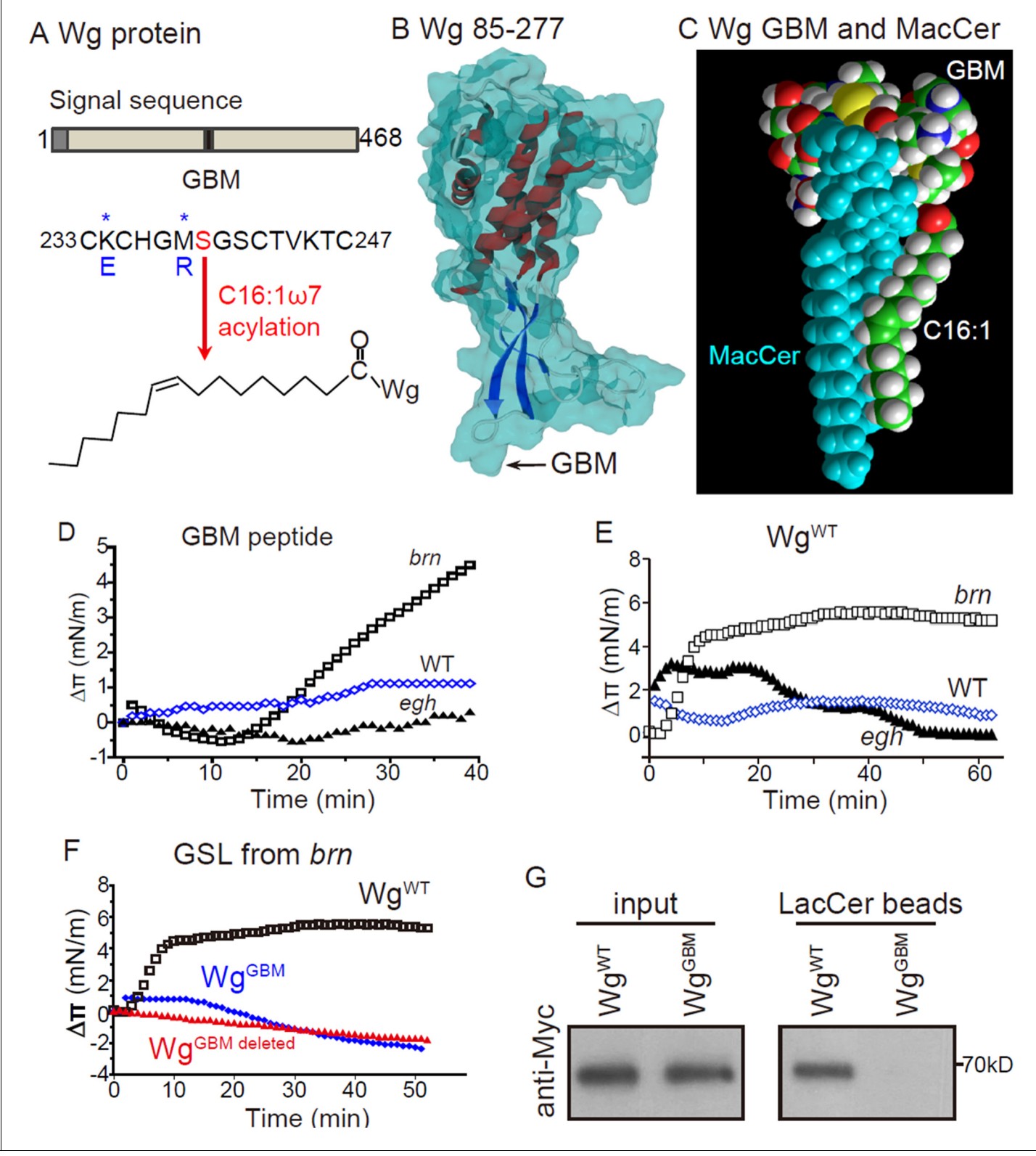

**Figure 8.** Wg has a functional MacCer-binding domain (**A**) Schematic representation of Wg protein and amino acid sequence of a putative GSL-binding motif (GBM) and mutated GBM with asterisks denoting the mutated amino acids.(**B**) A 3D model of Wg residues 85–277 with the location of the GBM, which exposes on the protein surface. (**C**) Energy-minimized model of the acylated (C16:1 ω7) GBM in Wg binding with MacCer. (**D and E**) Surface pressure change ($\Delta\pi$) by the addition of Wg[233-247] peptide (GBM peptide; **D**) and full-length Wg produced by in vitro translation system (**E**) on a

*Figure 8 continued on next page*

*Figure 8 continued*
monolayer of GSLs purified from wild type, *egh*[62d18] and *brn*[1.6P6] larvae. GBM peptide and full-length Wg showed specific high affinity to MacCer enriched GSLs. (F) Surface pressure change (Δπ) by the addition of Myc-tagged wild-type Wg (Wg[WT]) and Wg with GBM deleted (Wg[GBM deleted]) or mutated (Wg[GBM]) on a monolayer of GSLs purified from *brn*[1.6P6] larvae. All surface pressure measurements were performed in triplicate and a representative curve is shown. (G) Wild-type Wg (Wg[WT]) but not Wg with mutated GBM (Wg[GBM]) was pulled-down by LacCer-beads. Five biological repeats were carried out for the pull-down assay and a representative blot is shown.
DOI: https://doi.org/10.7554/eLife.38183.025
The following figure supplement is available for figure 8:

**Figure supplement 1.** Additional monolayer data and GSL profiles of *egh* and *brn* mutants.
DOI: https://doi.org/10.7554/eLife.38183.026

showed normal postsynaptic differentiation with no increase in ghost bouton number as well as normal SSR ultrastructure (*Figure 7*), suggesting a specific regulation of presynaptic rather than postsynaptic Wg activity by MacCer. A recent study showed that at NMJs, Wg is secreted from glial cells as well as presynaptic motor neurons, and that this glia-derived Wg controls the assembly of postsynaptic machinery at NMJs without affecting the bouton number (*Kerr et al., 2014*). The normal postsynaptic differentiation in *egh* mutants may be due to normal postsynaptic Wg signaling triggered by glia-derived Wg. This postulation is supported by the observation that manipulations of glial MacCer by genetic means did not affect NMJ growth (*Figure 2—figure supplement 2*).

How does MacCer affect Wg signaling at the molecular and cellular levels? Wg associates with lipid rafts via its acylation (*Zhai et al., 2004*), enabling its access to the raft-component MacCer. Upon lipid raft disruption by sterol depletion, Wg de-localizes from the lipid rafts (detergent-resistant, cholesterol-rich membrane microdomains) revealed by biochemical fractionation (*Zhai et al., 2004*). We envisage that delocalization of Wg does not allow its binding to MacCer, compromises its transport to the NMJ terminals (*Figure 5—figure supplement 1*), and in turn restricts NMJ growth (*Figure 4*). This notion is supported by colocalization of MacCer and Wg on specific membrane domains of presynaptic organelles (*Figure 6*) and direct MacCer-Wg binding (*Figure 8*). As MacCer is both required and sufficient for normal Wg expression at NMJ boutons, but not vice versa (*Figures 6* and *9*), we propose that MacCer acts as an adaptor for Wg at the presynaptic recycling endosomes (*Figure 9*). Our in silico data combined with biochemical and genetic analysis of GSL-Wg interaction further showed that the 233 – 247 domain of Wg, which is acylated at Ser-239, is a functional GBM that mediates high affinity binding to MacCer. Although we could not exclude the possibility that Wg GBM might have other function independent of MacCer binding, our in vivo data support that the GBM plays important roles in Wg-MacCer colocalization, Wg signaling and NMJ development (*Figure 9*).

The association of Wg with membrane rafts requires the acylation at Ser-239 mediated by Porcupine, a process conserved from *Drosophila* to vertebrates (*Galli et al., 2007*; *Herr and Basler, 2012*; *Zhai et al., 2004*). Importantly, the acylation by Porcupine is essential for the recognition of Wg by Evi thereby affecting the transport and secretion of Wg through recycling endosomes and exosomes (*Herr and Basler, 2012*; *Koles et al., 2012*; *Korkut et al., 2009*). Furthermore, both Porcupine and Evi are known to regulate the synaptic level of Wg and the signaling activity (*Korkut et al., 2009*; *Packard et al., 2002*). In the present study, we showed a reduction in Wg level at NMJ terminals but an increase in the neuropil of the ventral nerve cord of *egh* mutants (*Figure 6* and *Figure 6—figure supplement 2*), similar to that in *evi* mutants with impaired Wg transport and secretion (*Korkut et al., 2009*). These findings suggest that the transport and trafficking of Wg via endosomes may be compromised when MacCer is deficient. The detailed mechanism by which MacCer regulates Wg transport and trafficking remains to be further investigated. As Wnt signaling, which is widely involved in various developmental processes and diseases including neurological and psychological diseases, is evolutionarily conserved (*Herr et al., 2012*; *Korkut and Budnik, 2009*; *Salinas, 2012*), our finding that MacCer facilitates Wg signaling transduction suggests a new target for intervening Wnt signaling associated pathogenesis.

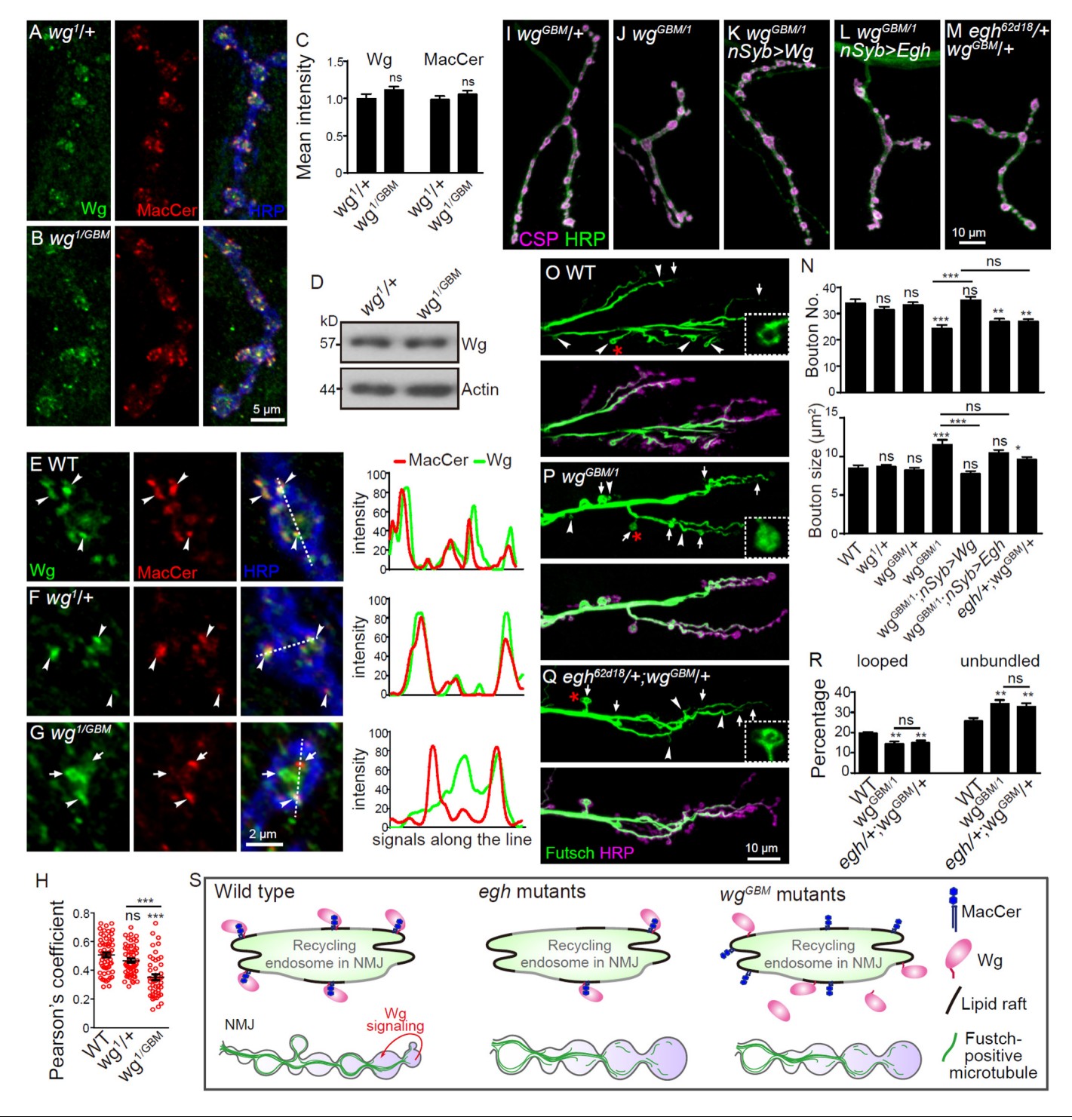

**Figure 9.** Wg GBM is required for Wg-MacCer colocalization and NMJ growth. (**A and B**) Representative images of NMJ4 co-labeled with anti-Wg (green), anti-MacCer (red) and anti-HRP (blue) in $wg^1/+$ (**A**) and $wg^{GBM}/wg^1$ mutants (**B**). (**C**) Quantification of Wg and MacCer intensities normalized to HRP intensities in different genotypes. $n \geq 15$ larvae; ns, no significance by one-way ANOVA with Tukey *post hoc* tests; error bars: s.e.m. (**D**) Western results of Wg and Actin from 3rd instar larvae of $wg^1/+$ and $wg^{GBM}/wg^1$ mutants. Actin was used as a loading control. Western blots were performed in triplicate and a representative image is shown. (**E–G**) Representative images of NMJ4 boutons triple-labeled with anti-Wg (green), anti-MacCer (red) and anti-HRP (blue) in wild type (**E**), $wg^1/+$ (**F**) and $wg^{GBM}/wg^1$ mutants (**G**). Arrowheads indicate puncta positive for both MacCer and Wg signals; arrows indicate puncta without obvious overlap of MacCer and Wg. Images were processed by deconvolution. Scale bar: 2 μm. Plot profiles of relative intensity of MacCer and Wg along the dashed lines were shown. (**H**) Pearson's coefficients of colocalization between MacCer and Wg. n = 58/10, 57/8, *Figure 9 continued on next page*

*Figure 9 continued*

and 46/8 (from left to right, boutons/animals). ns, no significance, ***p<0.001 by one-way ANOVA with Tukey *post hoc* tests; error bars: s.e.m. (*I–M*) Representative images of NMJ4 co-stained with anti-HRP (green) and anti-CSP (magenta) in $wg^{GBM}$/+ (**I**), $wg^{GBM}$/$wg^1$ (**J**), UAS-Wg-HA, $wg^{GBM}$/$wg^1$; nSyb-Gal4/+ (**K**), UAS-Egh, $wg^{GBM}$/$wg^1$; nSyb-Gal4/+ (**L**), and $egh^{62d18}$/+; $wg^{GBM}$/+ (**M**). Scale bar: 10 µm. (**N**) Quantifications of bouton number and bouton size of NMJs in different genotypes. $n \geq 12$ larvae; ns, no significance, *p<0.05; **p<0.01; ***p<0.001 by one-way ANOVA with Tukey *post hoc* tests, error bars: s.e.m. (*O–Q*) Representative images of NMJ6/7 labeled with anti-Futsch (green) and anti-HRP (magenta) in wild type (**O**), $wg^{GBM}$/$wg^1$ (**P**), and $egh^{62d18}$/+; $wg^{GBM}$/+ transheterozygotes (**Q**). Insets show higher magnification images of Futsch staining of a single bouton (asterisk). Arrows indicate boutons displaying unbundled Futsch; arrowheads denote boutons with Futsch loops. Scale bar: 10 µm. (**R**) Quantifications of percentage of Futsch loops and unbundled Futsch staining in different genotypes. $n \geq 12$ larvae; ns, no significance, **p<0.01 by one-way ANOVA with Tukey *post hoc* tests, error bars: s.e.m. (**S**) A schematic presentation of the role for MacCer in promoting synapse growth via interacting with Wg in lipid rafts. The presynaptic Wg signaling is denoted by a curved arrow.

DOI: https://doi.org/10.7554/eLife.38183.027

The following source data is available for figure 9:

**Source data 1.** Numerical data for the statistical graphs.

DOI: https://doi.org/10.7554/eLife.38183.028

# Materials and methods

## Key resources table

| Reagent type (species) or resource | Designation | Source or reference | Identifiers | Additional information |
|---|---|---|---|---|
| Genetic reagent (*D. melanogaster*) | $wg^{GBM}$ | this paper | NA | This allele carries the point mutation of K234E and M238R in Wg |
| Genetic reagent (*D. melanogaster*) | $wg^1$ | doi: 10.1523/Jneurosci.3714–13.2014 | RRID: BDSC_2978 | |
| Genetic reagent (*D. melanogaster*) | $wg^{CX4}$, $wg^{I-17}$ | doi: 10.1016/S0092-8674 (02)01047–4 | RRID: BDSC_2980 | |
| Genetic reagent (*D. melanogaster*) | UAS-Wg-HA(6C) | doi: 10.1016/S0092-8674 (02)01047–4 | RRID: BDSC_5918 | |
| Genetic reagent (*D. melanogaster*) | UAS-Wg-HA(3C) | doi: 10.1016/S0092-8674 (02)01047–4 | RRID: DGGR_108488 | |
| Genetic reagent (*D. melanogaster*) | $egh^7$ | PMID: 9012507 | RRID: BDSC_3902 | |
| Genetic reagent (*D. melanogaster*) | $egh^{62d18}$ | PMID: 9012507 | NA | Dr. Stephen M. Cohen (National University of Singapore) |
| Genetic reagent (*D. melanogaster*) | UAS-Egh-Myc | doi: 10.1074/jbc.C400571200 | NA | |
| Genetic reagent (*D. melanogaster*) | $brn^{fs107}$ | PMID: 1483386 | RRID: BDSC_4303 | |
| Genetic reagent (*D. melanogaster*) | $brn^{1.6P6}$ | PMID: 1483386 | RRID: BDSC_50762 | |
| Genetic reagent (*D. melanogaster*) | UAS-Brn | doi: 10.1016/j.ydbio.2007.04.013 | NA | |
| Genetic reagent (*D. melanogaster*) | UAS-brn-RNAi | doi: 10.1038/nmeth | RRID: BDSC_55386 | |

*Continued on next page*

*Continued*

| Reagent type (species) or resource | Designation | Source or reference | Identifiers | Additional information |
|---|---|---|---|---|
| Genetic reagent (*D. melanogaster*) | arr$^{k08131}$ | doi: 10.1534/ genetics. 104.026427 | RRID: BDSC_665 | |
| Genetic reagent (*D. melanogaster*) | arr$^2$ | doi: 10.1038/ 35035110 | RRID: BDSC_3087 | |
| Genetic reagent (*D. melanogaster*) | evi$^2$ | doi: 10.1074/ jbc.M112. 342667 | NA | Dr. Vivian Budnik (University Massachusetts Medical School) |
| Genetic reagent (*D. melanogaster*) | lace$^2$ | doi: 10.1128/ MCB.19. 10.7276 | RRID: BDSC_3159 | |
| Genetic reagent (*D. melanogaster*) | lace$^{k05305}$ | doi: 10.1128/ MCB.19. 10.7276 | RRID: BDSC_12176 | |
| Genetic reagent (*D. melanogaster*) | Df(2L) Exel7063 | NA | RRID: BDSC_7831 | |
| Genetic reagent (*D. melanogaster*) | schlank$^{G0489}$ | doi:10.1038/ emboj. 2009.305 | RRID: DGGR_111936 | |
| Genetic reagent (*D. melanogaster*) | schlank$^{G0061}$ | doi:10.1038/ emboj. 2009.305 | RRID: BDSC_11665 | |
| Genetic reagent (*D. melanogaster*) | Sk2$^{KG05894}$ | doi: 10.1194/ jlr.M300005- JLR200 | RRID: BDSC_14133 | |
| Genetic reagent (*D. melanogaster*) | Df(3L) BSC671 | NA | RRID: BDSC_26523 | |
| Genetic reagent (*D. melanogaster*) | UAS-ManII-GFP | doi:10.1016/ j.cell. 2007.06.032 | RRID: BDSC_65248 | Dr. Yuh Nung Jan (University of California, San Francisco) |
| Genetic reagent (*D. melanogaster*) | UAS-Spin-GFP | doi: S089662 7302010140 | RRID: BDSC_39668 | Dr. Graeme Davis (University of California, San Francisco) |
| Genetic reagent (*D. melanogaster*) | UAS-YFP-Rab5 | doi: 10.1534/ genetics.106. 066761 | RRID: BDSC_24616 | |
| Genetic reagent (*D. melanogaster*) | nSyb-Gal4 | NA | RRID: BDSC_51635 | |
| Genetic reagent (*D. melanogaster*) | OK6-Gal4 | NA | RRID: BDSC_64199 | |
| Genetic reagent (*D. melanogaster*) | C57-Gal4 | NA | RRID: BDSC_32556 | |
| Genetic reagent (*D. melanogaster*) | Repo-Gal4 | NA | RRID: BDSC_7415 | |
| Genetic reagent (*D. melanogaster*) | act-Cas9 | NA | RRID: BDSC_54590 | |
| Antibody | Mouse IgM anti-MacCer | doi: 10.1074/ jbc.C4005 71200 | NA | IHC (1:1) |
| Antibody | Mouse anti-CSP | DSHB Cat. #: 6D6 | RRID: AB_528183 | IHC (1:300) |
| Antibody | Mouse anti-Wg | DSHB Cat. #: 4D4 | RRID: AB_528512 | IHC (1:10), WB (1:50) |

*Continued on next page*

*Continued*

| Reagent type (species) or resource | Designation | Source or reference | Identifiers | Additional information |
|---|---|---|---|---|
| Antibody | Mouse anti-Futsch | DSHB Cat. #: 22C10 | RRID: AB_528403 | IHC (1:50) |
| Antibody | Mouse anti-DLG | DSHB Cat. #: 4F3 | RRID: AB_528203 | IHC (1:500) |
| Antibody | Mouse anti-GluRIIA | DSHB Cat. #: 8B4D2 | RRID: AB_528269 | IHC (1:200) |
| Antibody | Mouse anti-Fas II | DSHB Cat. #: 1D4 | RRID: AB_528235 | IHC (1:50) |
| Antibody | Mouse anti-Syx1A | DSHB Cat. #: 8C3 | RRID: AB_528484 | IHC (1:20), WB (1:1000) |
| Antibody | Mouse anti-dFz2 | DSHB Cat. #:12A7 | RRID: AB_528257 | IHC (1:5) |
| Antibody | Mouse anti-Rab11 | BD Biosciences Cat. #: 610656 | RRID: AB_397983 | IHC (1:50) |
| Antibody | Mouse anti-HA | MBL International Cat. #: M180 | RRID: AB_10951811 | IHC (1:1000) |
| Antibody | Mouse anti-Myc | CWBIO Cat. #: CW0259 | | IHC (1:300), WB (1:1000) |
| Antibody | Mouse anti-Myc | MBL International Cat. #: M192 | RRID: AB_11160947 | IHC (1:300), WB (1:1000) |
| Antibody | Rat anti-GFP | MBL International Cat. #: D153 | RRID: AB_591820 | IHC (1:200) |
| Antibody | fluorescence-conjugated anti-HRP | Jackson Immuno Research | RRID: AB_2314647 | IHC (1:200) |
| Antibody | Mouse anti-β-actin | Millipore Bioscience Research | RRID: AB_2223041 | WB (1:50000) |
| Chemical compound, drug | D,L-threo-PDMP | Matreya | Cat.#: 1719 | 0.5 mg/ml |
| Chemical compound, drug | filipin III | Cayman | 70440 | 50 µg/ml |
| Chemical compound, drug | MβCD | Sigma-Aldrich | C4555 | 20 mM |
| Chemical compound, drug | Lactosyl ceramide | Matreya | Cat.#: 1500 | |
| Peptide, recombinant protein | GBM | this paper | | CKCHGMSGSCTVKTC |
| Peptide, recombinant protein | GBM$^{mut}$ | this paper | | CECHGRSGSCTVKTC |
| Recombinant DNA reagent | pcDNA3.1-Myc-Wg | Invitrogen | V790-20 | |
| Recombinant DNA reagent | U6b-sgRNA | Addgene | 65956 | |
| Recombinant DNA reagent | pBluescript SK (-) | Stratagene | 212206 | |
| Commercial assay or kit | TNT T6 Quick Coupled Transcription/Translation System | Promega | Cat.#: L1171 | |
| Commercial assay or kit | LacCer-coated beads | Echelon | Cat.#: P-B0LC | |

*Continued on next page*

*Continued*

| Reagent type (species) or resource | Designation | Source or reference | Identifiers | Additional information |
|---|---|---|---|---|
| Commercial assay or kit | Control beads | Echelon | Cat.#: P-B000 | |
| Commercial assay or kit | HPTLC Silica Gel (aluminium plates) | Merck-Millipore | Cat.#: 105547 | |
| Software, algorithm | Hyperchem software | Hypercube | | |

## *Drosophila* strains and genetics

Flies were cultured on standard cornmeal media at 25°C. $w^{1118}$ was used as the wild type control, unless otherwise indicated. $egh^{62d18}$, $brn^{1.6P6}$, *UAS-Egh-Myc* (*Wandall et al., 2005*) and *UAS-Brn* (*Chen et al., 2007*) were provided by S. M. Cohen and S. Pizette. $evi^2$ was from V. Budnik (*Korkut et al., 2009*). *UAS-ManII-GFP* was from Y. N. Jan (*Ye et al., 2007*). *UAS-Spin-GFP* was from G. Davis (*Sweeney and Davis, 2002*). The following fly lines were obtained from the Bloomington Stock Center: *Df(2L)Exel7063*, *Df(3L)BSC671*, $Sk2^{KG05894}$, $lace^2$, $lace^{k05305}$, $schlank^{G0489}$, $schlank^{G0061}$, $egh^7$, $brn^{fs107}$, $arr^2$, $arr^{k08131}$, $wg^{CX4}$ ($wg^{[I-17]}$), $wg^1$, $rab11^{93Bi}$, *nSyb-Gal4*, *OK6-Gal4*, *C57-Gal4*, *Repo-Gal4*, *UAS-Wg-HA*, *UAS-brn-RNAi*, *UAS-YFP-Rab5* and *act-Cas9*. For fly strains used in the genetic screen, *UAS-GalNAc-TA* and *UAS-GalNAc-TB* were provided by S. M. Cohen (*Chen et al., 2007*). $desat1^{10A}$, $desat1^{11A}$, $desat1^{119A}$ and $desat1^{EY07679}$ were provided by E. Hafen (*Köhler et al., 2009*). Other fly lines were obtained from the Bloomington Stock Center, Kyoto Stock Center and the Tsinghua *Drosophila* RNAi Center.

## Site-directed mutation in *wg*

We generated double site-directed point mutations at K234E and M238R in Wg to test the function of GBM. The CRISPR/Cas9-mediated targeted mutagenesis of *wg* (*CG4889*) from $w^{1118}$ was performed largely according to previously published homology-directed repair procedures (*Gratz et al., 2014*; *Port et al., 2014*) at Fungene Biotech (http://www.fgbiotech.com). sgRNA (synthetic guide RNA) targets were designed with CRISPR Optimal Target Finder (*Gratz et al., 2014*). Two sgRNAs recognized the region containing the mutations (gcgacaggagtgcaaatgcca*tgg* and gcaaatgccatggcatgtc*cgg*; the last three bases *ngg* denotes proto-spacer adjacent motif) were cloned into U6b-sgRNA vector (*Ren et al., 2013*). For donor vector construction, a 1.8 kb *wg* gene region containing the site-directed mutation was cloned into pBluescript SK (-) vector. The donor vector with the site-directed mutation and the sgRNA vector were injected into *act-Cas9* transgenic embryos. PCR sequencing was performed to identify if the offspring flies carried the designed mutation.

## Immunochemical analysis

Immunostaining of larval preparations was performed as previously described (*Liu et al., 2014*). For most antibody staining, specimens were dissected in $Ca^{2+}$-free HL3 saline, fixed in 4% paraformaldehyde for 30 min, and washed in 0.2% Triton X-100 in PBS. For MacCer staining, specimens were fixed in 4% paraformaldehyde at 4°C for 60 min and permeabilized with 0.1% Triton X-100 in cold PBS, incubated with undiluted anti-MacCer hybridoma supernatants for 18 hr at 4°C and detected with Fluor 568 conjugated goat anti-mouse IgM (1:1000; Invitrogen). For GluRIIA staining, specimens were fixed in cold methanol for 5 min. We used the following antibodies: mouse anti-cysteine string protein (CSP) (1:300; 6D6 from DSHB), mouse anti-Wg (1:10; 4D4 from DSHB), mouse anti-Futsch (1:50; 22C10 from DSHB), mouse anti-Rab11 (1:50; BD Biosciences), mouse anti-DLG (1:500; 4F3 from DSHB), mouse anti-GluRIIA (1:200; 8B4D2 from DSHB), mouse anti-Fas II (1:50; 1D4 from DSHB), mouse anti-Syx1A (1:20; 8C3 from DSHB), mouse anti-dFz2 (1:5; 12A7 from DSHB), mouse anti-HA (1:1000; MBL International), mouse anti-Myc (1:300; CWBIO, MBL International), rat anti-GFP (1:200; MBL International), and fluorescence-conjugated anti-horseradish peroxidase (HRP) (1:200; Jackson ImmunoResearch). Primary antibodies were visualized using corresponding secondary antibodies conjugated to Fluor 488, Cy3 (both at 1:1000; Invitrogen) or DyLight 649 (1:500; Jackson ImmunoResearch).

Western analysis was performed as previously described (*Liu et al., 2014*). Briefly, 3rd instar larval brains were dissected in cold PBS and homogenized in RIPA buffer (50 mM Tris-HCl at pH 7.4, 150 mM NaCl, 0.1% SDS, 1% NP-40) with proteinase inhibitors Set I (Calbiochemistry) on ice. Samples were then subjected to SDS-PAGE and immunoblotting according to standard procedures. The following antibodies were used: mouse anti-Wg (1:50; 4D4 from DSHB), mouse anti-Syx1A (1:1000; 8C3 from DSHB), and mouse anti-actin (1:50000; Millipore Bioscience Research Reagents). All Western blotting were performed at least for three independent biological replicates.

## Image collecting and statistical analysis

Image collecting was carried out as previously described (*Liu et al., 2014*). NMJ images were collected with an Olympus Fluoview FV1000 confocal microscope using a 40x/1.42 NA or 60 x/1.42 NA oil objective and FV10-ASW software, or with a Leica confocal microscope using a 40x/1.25 NA oil objective and LAS AF software. All images of muscle 4 or muscle 6/7 glutamatergic type Ib NMJ of abdominal segments A2 to A4 for a specific experiment were captured using identical settings for statistical comparison among different genotypes. High resolution confocal images were processed with the deconvolution software AutoQuant X2 (Media Cybernetics). Brightness, contrast, and color were adjusted using Photoshop CS5 (Adobe). NMJ morphological features were quantified using ImageJ (National Institutes of Health, Bethesda, MD).

For quantification analysis of NMJ growth, individual boutons were defined according to the discrete staining signal of anti-CSP. Satellite boutons were defined as extensions of five or fewer small boutons emanating from the main branch of the NMJ terminals. For quantification of bouton sizes, synaptic areas ($\mu m^2$) were measured by assessing HRP-positive boutons, and normalized to bouton numbers. Ghost boutons were defined as boutons positive for HRP staining but negative for DLG staining. NMJ bouton number and size of *schlank* mutants were normalized to muscle surface area. For quantification of protein levels at NMJ, staining intensities of each protein were measured within HRP-positive NMJs using ImageJ. For colocalization analysis, single slices were analyzed using WCIF-ImageJ 'Colocalization Analysis' within HRP-labeled synaptic boutons. Statistical comparisons were performed using GraphPad Prism 6. Data of NMJ features are expressed as mean ± standard error of the mean (s.e.m.). Statistical significance between each genotype and the controls was determined by two tailed Student's *t* test, whereas multiple comparisons between genotypes were determined by one-way ANOVA with a Tukey *post hoc* test. Asterisks above a column indicate comparisons of a specific genotype to wild type (denoted WT) or genetic control (denoted CT), whereas asterisks above a bracket denote comparisons between two specific genotypes. ns denotes $p > 0.05$; * indicates $p < 0.05$; ** denotes $p < 0.01$; *** indicates $p < 0.001$.

## Drug treatment of larvae

Larvae of different genotypes were raised on vehicle- or drug-containing media from egg hatching. D,L-threo-PDMP (Matreya) and filipin III (Cayman) were dissolved in DMSO and then individually added to standard media at specific concentrations. All the treatments used DMSO vehicle at a concentration of 0.5%. MβCD (Sigma-Aldrich) was resolved in standard media at a final concentration of 20 mM.

## Molecular simulation

Docking of Wg onto MacCer was performed with the Hyperchem 8.0 program as described previously (*Fantini and Yahi, 2011*). Homology modeling of Wg fragment of 85 – 277 amino acids was based on the published crystal structure of XWnt8 (*Janda et al., 2012*).

## Lipid–protein interaction assay

Synthetic peptides of GBM (95%) were purchased from Sangon Biotech. Myc-tagged full-length Wg were produced by an in vitro translation kit containing rabbit reticulocyte lysates (Promega). pCDNA-Myc-Wg was expressed in the translation system at 30°C for 90 min. Myc-Wg was immunoprecipitated with anti-Myc affinity resin (Millipore), eluted with a Myc-peptide solution (Sigma), and concentrated using a 30kD filter column (Amicon).

GSL–protein interactions were determined with the Langmuir-film balance technique as previously described (*Di Scala et al., 2014*). It was monitored by surface pressure with a fully automated

microtensiometer (µTrough SX; Kibron Inc.). All experiments were performed at 20 ± 1°C. Monomolecular films of GSLs were spread on pure water subphases (800 µl) from chloroform/methanol (1:1 vol/vol). The initial surface pressure of the monolayers was 12–15 mN/m. The accuracy of our experimental conditions was ±0.25 mN/m. Real-time measurements kinetically followed the increase in the surface pressure after injecting the peptide or protein (final concentration of 10 µM) into the aqueous phase underneath the GSL monolayer until equilibrium was reached. The data were analyzed with the FilmWareX program (version 3.57; Kibron Inc.).

Lipid–protein pull-down assay was modified following a previous protocol (*Kunisaki et al., 2006*). Wild-type and mutated Myc-Wg were produced by an in vitro translation kit containing rabbit reticulocyte lysates (Promega). The reticulocyte lysates were suspended in 400 µL lipid-binding/washing buffer (10 mM HEPES, pH 7.4, 150 mM NaCl, 0.35% NP-40) and incubated with 20 µL slurry of control or LacCer-coated beads (Echelon) at 4°C for 3 hr under rotary agitation. After extensive washing with lipid-binding/washing buffer, the bound proteins were eluted from beads with 2 × Laemmli sample buffer and subjected to Western analysis with mouse anti-Myc (1:1000; CWBIO).

### GSL extraction and analysis

GSL extraction was performed as described (*Hamel et al., 2010*; *Wandall et al., 2005*). 3rd instar larvae were homogenized in 1.5 ml solvent A (2-isopropanol/hexane/water; 55:25:20 vol/vol/vol). The homogenate was centrifuged at 2,000 rpm and the supernatant was collected. This step was repeated with 1.5 ml solvent B (chloroform/methanol; 1:1 vol/vol), 1.5 ml solvent A, and finally 1.5 ml solvent B. The four solvent extracts were combined, evaporated under a nitrogen flux, and re-suspended in chloroform/methanol (2:1 vol/vol) at a lipid concentration of 1 mg/ml. The extracts were again evaporated, then re-suspended in 5 ml methanol containing 0.1 M NaOH, and incubated for 1 hr at 37°C under agitation to remove most glycerolipids. The samples were evaporated and re-extracted in chloroform/methanol (2:1 vol/vol). Neutral GSLs were finally purified on a column (DEAE-Sephadex A-25; Sigma-Aldrich) and eluted with chloroform/methanol/water (30:60:8 vol/vol/vol). GSL extracts were analyzed by high performance thin layer chromatography (HPTLC) using silica gel 60 HPTLC plates (Merck) in chloroform/methanol/water (60:35:8 vol/vol/vol). The HPTLC plates were sprayed with orcinol and heated at 110°C for GSL detection. GSLs bands were identified by their mobility relative to GSL standards (Matreya).

### Electron microscopy

Electron microscopy (EM) was performed as described (*Liu et al., 2014*). Briefly, dissected 3rd instar larvae were fixed overnight at 4°C in 2% glutaraldehyde plus 2% paraformaldehyde in 0.1 M cacodylate buffer (pH 7.4), and post-fixed in 1% $OsO_4$ in cacodylate buffer for 90 min at room temperature. Samples were stained in saturated aqueous uranyl acetate for 1 hr, dehydrated in a graded acetone series and embedded in Spurr resin (Electron Microscopy Sciences). Type 1b boutons from NMJ 6/7 in abdominal segments A2 to A4 were serially sectioned with a Leica UC6 ultramicrotome, stained with uranyl acetate and Sato's lead, and observed using a JEOL 1400 electron microscope. The number of synaptic vesicles within a 200 nm radius of the active zones (AZs) in *egh* mutants and wild type was quantified. Other features were quantified as previously described (*Packard et al., 2002*).

## Acknowledgements

We thank S M Cohen, S Pizette, V Budnik, E Hafen, Y N Jan and G Davis, Tsinghua *Drosophila* RNAi Center and the Bloomington Stock Center for fly stocks. We thank H Chahinian for his help in the purification and analysis of GSLs and W Li for her help in Wg production and site-directed mutation in *wg*. We thank Z Liu and R K Yu for discussion and comments on the manuscript. This work was supported by grants from the Ministry of Science and Technology of the people's Republic of China; and the National Science Foundation of China (NSFC). The authors declare no competing financial interests

## Additional information

### Funding

| Funder | Grant reference number | Author |
|---|---|---|
| Ministry of Science and Technology of the People's Republic of China | 2014CB942803 | Yong Q Zhang |
| Ministry of Science and Technology of the People's Republic of China | 2016YFA0501002 | Yong Q Zhang |
| National Natural Science Foundation of China | 31490590 | Yong Q Zhang |
| National Natural Science Foundation of China | 31500846 | Yan Huang |

The funders had no role in study design, data collection and interpretation, or the decision to submit the work for publication.

### Author contributions

Yan Huang, Conceptualization, Funding acquisition, Investigation, Writing—original draft, Project administration, Writing—review and editing; Sheng Huang, Coralie Di Scala, Qifu Wang, Investigation; Hans H Wandall, Resources; Jacques Fantini, Supervision, Investigation, Writing—original draft; Yong Q Zhang, Conceptualization, Supervision, Funding acquisition, Validation, Writing—review and editing

### Author ORCIDs

Yan Huang [ID] http://orcid.org/0000-0003-3833-9248
Coralie Di Scala [ID] http://orcid.org/0000-0003-0655-7056
Hans H Wandall [ID] http://orcid.org/0000-0003-0240-9232
Jacques Fantini [ID] http://orcid.org/0000-0001-8653-5521
Yong Q Zhang [ID] https://orcid.org/0000-0003-0581-4882

### Decision letter and Author response

Decision letter https://doi.org/10.7554/eLife.38183.032
Author response https://doi.org/10.7554/eLife.38183.033

## Additional files

### Supplementary files

• Supplementary File 1. Genetic screen for NMJ morphology by manipulating selected genes involved in synthesis and turnover of membrane lipids.
DOI: https://doi.org/10.7554/eLife.38183.029

• Transparent reporting form
DOI: https://doi.org/10.7554/eLife.38183.030

All data generated or analysed during this study are included in the manuscript and supporting files.

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
