## [Decision Letter]

Thank you for submitting your work entitled "The glycosphingolipid MacCer promotes synaptic bouton formation in *Drosophila* by interacting with Wnt" for consideration by *eLife*. Your article has been reviewed by three peer reviewers, one of whom is a member of our Board of Reviewing Editors, and the evaluation has been overseen by a Reviewing Editor and a Senior Editor. The following individuals involved in the review of your submission have agreed to reveal their identity: Thomas Lloyd (Reviewer #3).

The reviewers have discussed the reviews with one another and the Reviewing Editor has drafted this decision.

We are concerned that the revisions required will take significantly more than two months. We are therefore turning down the manuscript. We, however, encourage re-submission when you feel you are able to address these substantive concerns.

Summary:

Huang et al. show that glycosphigolipid (GSL) MacCer regulates *Drosophila* neuromuscular junction development by genetically manipulating enzymes in MacCer production. These results are internally consistent, and they show that reduction of MacCer correlates with fewer and larger synaptic boutons, a phenotype that is reminiscent of Wg mutants. The authors go on to show that the Wg and GSL pathways genetically interact and they provide some evidence that MacCer binds Wg via lipid rafts, although we found these results at the moment less compelling. As you will see, the reviewers have made a list of constructive comments that we believe you will be able to address; most issues are of technical in nature. There is, however, one point that surfaced during the discussion that the reviewers felt was not yet sufficiently supported. This is the issue of showing conclusively that Wg binds to MacCer in vivo. The reviewers would like to see a second methodology where you show this. Given this is a central point in your paper we would encourage you to pursue experiments to address this.

*Reviewer #1:*

The manuscript "The glycosphingolipid MacCer promotes synaptic bouton formation in *Drosophila* by interacting with Wnt" by Huang et al., describes that MacCer interacts with Wg via a GSL-binding motif in Wg. The idea that Wg binds directly to MacCer in vivo is intriguing, but the presented data are not strong enough to make this point. Moreover, one of their main findings is also that MacCer is enriched in recycling endosomes, but there are no experiments made to uncover the biological significance of this finding.

A larger concern of mine is that the idea that sphinolipids regulate synaptic growth is not new and was described by the authors in a previous manuscript: "Acsl, the *Drosophila* ortholog of intellectual-disability-related ACSL4, inhibits synaptic growth by altered lipids" Huang et al., 2016. In that paper they showed that synaptic bouton formation is dependent on fatty acid membrane composition via Acsl function. Moreover, they showed that MacCer level were increased in Acsl mutant synapses. The authors are not properly placing the present paper in the context of their previous work are there is significant overlap with the current manuscript (e.g. in Figure 2 and Figure 3).

In summary, I think the current work shows potential new insights into lipid regulation at the synapse, but it clearly requires substantial additional experimental data to cover the consequence of MacCer function in recycling endosomes and in vivo data of MacCer binding to Wg and data showing what happens in wg GBM mutant animals especially in respect to MacCer.

Essential revisions:

- In the Materials and methods section, the authors stated that they used synapses form muscle 4 or 6/7 and abdominal segments A2 to A4. Synapses morphology vary significantly between these two muscle types and segments. It is important to separate out these data and show them per muscle and per segment.

- Figure 1: The authors used RNAi lines from the Vienna collection to screen for defects in NMJ morphology. There is no confirmation of knock down efficiency, target specify or off target effects. Two independent RNAi lines haven been used for each gene (schlank and lace), but the quantification of the phenotypes does not include both lines.

- Figure 2: The authors claim that egh function on NMJ growth is synapse specific. However, there are is no consistent quantification that could support this point regarding satellite boutons, bouton size and bouton nr.

- Figure 4: The authors did not show the effect of Filipin on MacCer levels.

- Figure 6 and Figure 6 supplements: The authors state that MacCer colocalize with Wg and Rab11 at the synapse, but they did not perform a Pearson Coefficient analysis to support these statements. The authors did not provide a co-localization analysis of MacCer with late endosomes, early endosomes and Golgi markers. Moreover, the authors should provide further evidence by using other more commonly used organelle markers. There is also no quantification of Western blot data to state that Wg levels in egh mutant brains are normal. Additional data on axonal transport would be more suitable to support the notion about Wg secretion into the synapse. Especially since it is known that egh and brn mutants show enlarged nerves (Dahlgaard et al., 2012). There are no representative images for MacCer and Fz2 fluorescent intensities in egh mutants and wild types in Figure 6—figure supplement 2C. Again, not all quantified genotypes show representative images in main Figure panel L.

- Figure 8: In silico predictions and physiochemical experiments suggest a direct binding of Wg to MacCer, but there is no evidence that this is also true in vivo. The authors made the wg GBM mutant and claim that this would affect lipid rafts via disrupted MacCer binding. However, the presented data are not very direct and the implication of MacCer levels and localization in wg GBM mutant animals is lacking.

*Reviewer #2:*

The authors screen a number of different genetic lipid synthesis mutants and identify abnormal NMJ development in animals where mutations inhibit sphingolipid metabolism. They then identify that enzymes that act sequentially on MacCer production have opposing effects on NMJ development; loss of Egh (that produces MacCer) produces fewer and larger boutons, while loss of Brn (that metabolizes MacCer to GlcNac-MacCer) produces more boutons including satellite boutons. They also conclude (largely based on sterol depletion) that presynaptic MacCer is biologically active within lipid rafts. The final section of the paper addresses the mechanism by which MacCer promotes synaptic growth (putatively from presynaptic lipid rafts). They describe that Wg and Egh genetically interact at the level of NMJ bouton number and size. They show abnormal Wg-signalling markers in Egh boutons, including that Wg and Rab11 colocalize with anti-MacCer labelling, Wg levels vary with Egh genotype, and use the presence/ absence of Wg-sensitive Futsch-loops to further test if MacCer facilitates Wg signalling. Finally, molecular modelling and in vitro analysis is used to support a model where Wg directly binds to MacCer.

Overall, while the topic is interesting, several data sets are over-interpreted, and controls are missing. The paper tries to make many points, but the data is lacking for some parts of their model. This is particularly true for the claims on SL synthesis (like Figure 1), as well as the concept that MacCer is active in lipid rafts and/ or endosomes. On the other hand, there is considerable data on the point that Egh (MacCer) genetically interacts with Wg and affects presynaptic Wg signalling – though again the manuscript falls short of convincingly proving a direct physical interaction (Figure 8).

One issue is that the first half of the paper (Figures 1-5) is limited to bouton numbers and bouton sizes. However, many molecular pathways converge to affect these variables and it is not possible to confidently assign genes (or drug treatments) to a single common molecular pathway. The authors should add additional readouts of synaptic development to see if their conclusions on genetic interactions stand up. There may be parts of this in the final figures showing Wg-related defects in Egh lines, but these analyses were not conducted for the majority of genetic interactions shown in Figures 1-5.

Related is that there are several experiments that use constitutive hypomorphs/ knock-out lines of widely expressed genes but interpret data to conclude that genes act presynaptically. For example, Figure 1 shows synaptic defects in constitutive mutants of Schlank, lace and Sk2, without genetically rescuing to show that they operate presynaptically. The Figure 5 genetic interaction between Wg and Egh uses many constitutive mutants. The authors need to better establish that all phenotypes are cell autonomous to motor neurons (RNAi and re-expression in mutant backgrounds with motor neuron-specific drivers).

This problem is amplified with the pharmacological treatments. Drugs that deplete sterols or GSL from egg hatching likely broadly impair development rather than specifically affect lipids in synapses as the authors interpret. At the very least control data is needed on larval size, time to pupate, and general tissue development, etc. The conclusion that sterol and SL depletion data show MacCer operating in lipid rafts is overinterpreted given the drug treatments will affect all cells in the animal from egg hatching and have multiple impacts at the level of neuronal function (not just disrupting presynaptic lipid rafts).

The final parts of the paper contain the more convincing data on a genetic interaction between Egh and Wg, including multiple signs of abnormal Wg singling in Egh mutants. However, it seems particularly critical to be sure of direct binding between Wg and MacCer and ideally the authors can show this via a second methodology since the manuscript has too little experimental data to prove this.

The concept that MacSer is specifically important and localised to endosomes also isn't fully established. If important, the authors should biochemically test this, for example via lipid MS of purified endosomes.

*Reviewer #3:*

Huang and colleagues demonstrate a novel role for the glycosphigolipid (GSL) MacCer in regulating *Drosophila* neuromuscular junction development. By genetically modulating enzymes up- and downstream of MacCer production, the authors nicely show that reduction of MacCer correlates with fewer and larger synaptic boutons. This NMJ phenotype is reminiscent of Wg mutants, and indeed, Wg and GSL pathway mutants genetically interact, and overexpression of Wg presynaptically rescues the MacCer deficient phenotype. The authors provide evidence that MacCer directly binds Wg via lipid rafts, and this underlies the role of this GSL in NMJ bouton development. The data provided are of high quality, and the conclusions are compelling.

The generation of the GBM Wingless mutant via CRISPR is an excellent new tool for the investigators to further explore the role of Wg-GSL interactions in Wg function. While I acknowledge that extensive characterization of this new allele is beyond the scope of this study, a few more experiments are necessary to test their hypothesis. First, the authors should examine bouton size in Wg[GBM/1] animals as they have done in previous figures to determine if this phenocopies mutants that cause reduction of MacCer. Second, the authors should show that these phenotypes can be rescued by presynaptic expression of wildtype Wg. Finally, it would be of great interest to determine if the Wg[GBM/1] phenotype is modulated by egh and brn genetic manipulations as was performed in prior figures. If the Wg[GBM/1] phenotype is epistatic to brn and egh as would be predicted by the model, an insensitivity of this GBM allele to MacCer levels would be strong genetic evidence that these GSLs are regulating synaptic development through Wg.

The alteration in MacCer levels in egh mutants are quite modest (Figure 3B). Is this because of the partial LOF nature of the alleles or due to alternative ways of generating MacCer? In this regard, the data provided (Figure 8—figure supplement 1C) don't convince me of their conclusion that "egh62d18 mutants contained chiefly GlcCer ".

In Figure2L, the authors show that brn[fs107/Y] mutants have a significant increase in bouton no. Presumably, this is compared with WT male larvae, though the control should be explicitly stated. What is the WT strain used, and is it the appropriate genetic background control? In Figure 2, which egh allele is used in the rescue experiments?

[Editors’ note: what now follows is the decision letter after the authors submitted for further consideration.]

Thank you for submitting your article "The Glycosphingolipid MacCer promotes synaptic bouton formation in *Drosophila* by interacting with Wnt" for consideration by *eLife*. Your article has been reviewed by K VijayRaghavan as the Senior Editor, a Reviewing Editor, and two reviewers.. The following individual involved in review of your submission has agreed to reveal his identity: Thomas Lloyd (Reviewer #2).

The reviewers have discussed the reviews with one another and the Reviewing Editor has drafted this decision to help you prepare a revised submission.

Summary:

Overall, the manuscript is much better and is nearly ready for acceptance. The data is now more convincing that presynaptic MacCer interacts with Wg and, in doing so, affects presynaptic Wg signaling and synaptic development. Figure 1 and Figure 2 now convincingly show that Egh and Brn act in motor neurons to oppositely affect synapse development. These data overlap partially with that in their 2016 Huang et al., publication. However, the extra genetics, in particular, that they analyze motor neuron-specific Egh and Brn expression and RNAi, do now establish the basal importance of MacCer that wasn't so clear before. The data also establish that Wg protein interacts directly with the MacCer lipid to complement the genetic interaction data.

Essential revisions:

The following essential revisions can be made by addressing them in the abstract, results and discussions. Many aspects, such as antibody specificity, are buried in the text for the reader to decipher as then go through. These and such concerns as given below should be addressed by re-writing in a more focused manner.

The data fails to convince on the lipid raft localized nature of this interaction, or precisely what this means. One issue is that we would expect that plasma membrane "lipid raft" domains also exist at these synapses, but yet the authors only detect internal signal with the MacCer antibody. The pharmacological approaches that disrupt sterols are broad and likely do much more than just affect lipid rafts. It feels like an over-interpretation to designate these to the same genetic pathway on the basis that the double manipulation is similar to sterol or MacCer loss alone; perhaps the phenotype is already as severe as it can be. Further, they interpret Figure 4 data to mean that lipid rafts have some impact on MacCer function, without further elucidating on what this might be. The most obvious functional impact is that lipid rafts concentrate MacCer to specific membrane domains, in turn facilitating binding to Wg protein. However, if so, why does MacCer staining remain punctate even after lipid rafts are disrupted by sterol depletion? If their model is correct, we would expect they see diffuse/ less intense/ less punctate MacCer labeling? A more detailed discussion of this should be added if it is indeed important or, ideally, they consider an in vitro system where lipid rafts +/- MacCer are easily detectable (for example on the surface of some mammalian cell lines).

The MacCer antibody is the only readout of MacCer levels (there is no direct analysis of lipid levels in the mutants). It would, therefore, be good if they provide data on antibody specificity. This could include ruling out the possibility that it detects a different molecule that also rises and falls with altered synaptic size; for example, whether MacCer staining remains normal with other manipulations that affect synapse structure (like in Wg mutants). Some of this may be already in the manuscript, but it would help if the issue is directly addressed given non-specific labeling would invalidate many of their conclusions. I also wonder about antibody accessibility when proteins interact with MacCer. Presumably, Wg and the antibody interact with the carbohydrate group of MacCer, so the MacCer antibody would detect less MacCer if it is already complexed with Wg (or other binding partners)? This seems like something to consider when interpreting colocalization data and when using the antibody to determine synaptic MacCer levels.

---

## [Author Response]

[Editors’ note: the author responses to the first round of peer review follow.]

Reviewer #1:

*The manuscript "The glycosphingolipid MacCer promotes synaptic bouton formation in Drosophila by interacting with Wnt" by Huang et al., describes that MacCer interacts with Wg via a GSL-binding motif in Wg. The idea that Wg binds directly to MacCer* in vivo *is intriguing, but the presented data are not strong enough to make this point. Moreover, one of their main findings is also that MacCer is enriched in recycling endosomes, but there are no experiments made to uncover the biological significance of this finding.*

To further support the binding of Wg and MacCer, we have added two independent lines of evidence. First, wild-type but not GBM mutation-carrying Wg binds LacCer (the vertebrate analog of MacCer) (Figure 8). Second, *wg^GBM^* mutation led to a decrease of co-localization between MacCer and Wg in NMJs (Figure 9; also see the response to the third point of this reviewer).

The biological function of MacCer in recycling endosomes is to facilitate membrane trafficking of Wg. MacCer enrichment in recycling endosomes is consistent with previous findings that lipid rafts/LacCer are enriched in recycling endosomes (Balasubramanian et al., 2007; Hortsch et al., 2010; Sabharanjak et al., 2002). The intracellular trafficking of lipid rafts from Golgi or early endosomes to plasma membrane via recycling endosomes is thought to impact trafficking of many raft-associated proteins (reviewed in Simons and Toomre, 2000). We have added these references on lipid rafts trafficking via recycling endosomes in in subsection “MacCer promotes NMJ bouton formation via Wg signaling” and subsection “MacCer promotes synaptic growth via presynaptic Wg signaling” in the revision (also see the response to the third point of this reviewer).

A larger concern of mine is that the idea that sphinolipids regulate synaptic growth is not new and was described by the authors in a previous manuscript: "Acsl, the Drosophila ortholog of intellectual-disability-related ACSL4, inhibits synaptic growth by altered lipids" Huang et al., 2016. In that paper they showed that synaptic bouton formation is dependent on fatty acid membrane composition via Acsl function. Moreover, they showed that MacCer level were increased in Acsl mutant synapses. The authors are not properly placing the present paper in the context of their previous work are there is significant overlap with the current manuscript (e.g. in Figure 2 and Figure 3).

We have previously reported that reducing MacCer level by *egh^62d18^* mutation or *brn*-overexpression in neurons lead to fewer and larger boutons as well as restrict the NMJ overgrowth in *Acsl* mutants (Huang et al., 2016). However, in that study, we did not dissect the mechanism by which MacCer regulates synapse growth. In the current study, we far extended our previous study and systematically and mechanistically analyze the role of MacCer in promoting NMJ growth. Specifically, we defined a presynaptic function of MacCer by cell-type specific manipulation of MacCer level and revealed a new regulatory mechanism that MacCer promotes NMJ growth via Wg signaling. We have discussed the present data in the context of the previous study in subsection “GSL synthases Egh and Brn bi-directionally regulatesNMJ growth 104 presynaptically” and subsection “MacCer promotes NMJ bouton formation via Wg signaling”.

*In summary, I think the current work shows potential new insights into lipid regulation at the synapse, but it clearly requires substantial additional experimental data to cover the consequence of MacCer function in recycling endosomes and* in vivo *data of MacCer binding to Wg and data showing what happens in wg GBM mutant animals especially in respect to MacCer.*

We have provided experimental evidence supporting that MacCer acts as an adaptor for Wg to be localized to lipid rafts/recycling endosomes. As mentioned above, it has been well established that intracellular trafficking of LacCer/lipid rafts is mediated primarily via recycling endosomes. The lipid raft trafficking via recycling endosomes plays a crucial role in raft-associated signaling transduction (reviewed in Simons and Toomre, 2000). Our finding that MacCer enriched in Rab11-positive recycling endosomes implicates trafficking of Wg via recycling endosomes.

To further support the binding of Wg and MacCer, we have added new data by a second methodology showing that wild-type Wg binds LacCer-beads whereas Wg carrying mutated GBM does not. These new data, in combination with our previous physical binding data (Figure 8), support a direct binding between MacCer and Wg via GBM. Furthermore, we have added new data that *wg^GBM^* mutation led to a decrease of co-localization between MacCer and Wg, without significantly affecting the synaptic level of MacCer (Figure 9). Our results together demonstrate that Wg GBM is critical for the interaction between Wg and MacCer, and that MacCer positively regulates Wg level, but not vice versa (also see the response to Figure 8 of this reviewer).

Essential revisions:- In the Materials and methods section, the authors stated that they used synapses form muscle 4 or 6/7 and abdominal segments A2 to A4. Synapses morphology vary significantly between these two muscle types and segments. It is important to separate out these data and show them per muscle and per segment.

The morphology of NMJ is indeed different between muscle 4 and muscle 6/7, but NMJ morphology in the same muscle from segments A2 to A4 is similar (Ballard et al., 2014; Chen and Ganetzky, 2012; Rodal et al., 2011; Zhao et al., 2015). We performed morphological analysis on NMJ4 as it is simple for quantification. NMJ6/7 terminal is larger with more boutons for electron microscopy and Futsch loop analysis. We indicated the use of specific NMJ for different experiments in Figure legends.

- Figure 1: The authors used RNAi lines from the Vienna collection to screen for defects in NMJ morphology. There is no confirmation of knock down efficiency, target specify or off target effects. Two independent RNAi lines haven been used for each gene (schlank and lace), but the quantification of the phenotypes does not include both lines.

We did not use any RNAi lines from the Vienna collection in this study. All the RNAi lines used in genetic screen (Supplementary file 1) are from Bloomington and Tsinghua RNAi center. We did not use RNAi lines for phenotypic analysis in Figure 1.

In Figure 2 and Figure 3, we used *brn* RNAi knockdown to study its cell type specific function. The phenotypes of *brn* RNAi was reminiscent of that in *brn* mutants (Figure 2, Figure 2—figure supplement 1, Figure 2—figure supplement 2 and Figure 3), suggesting an efficient knockdown.

- Figure 2: The authors claim that egh function on NMJ growth is synapse specific. However, there are is no consistent quantification that could support this point regarding satellite boutons, bouton size and bouton nr.

We showed that the function of *egh* on NMJ growth is pre-synapse specific rather than synapse specific. We have added quantifications on NMJ boutons of all genotypes in Figure Supplements.

- Figure 4: The authors did not show the effect of Filipin on MacCer levels.

We have added results of MacCer level at NMJ upon Filipin treatment.

- Figure 6 and Figure 6 supplements: The authors state that MacCer colocalize with Wg and Rab11 at the synapse, but they did not perform a Pearson Coefficient analysis to support these statements. The authors did not provide a co-localization analysis of MacCer with late endosomes, early endosomes and Golgi markers. Moreover, the authors should provide further evidence by using other more commonly used organelle markers. There is also no quantification of Western blot data to state that Wg levels in egh mutant brains are normal. Additional data on axonal transport would be more suitable to support the notion about Wg secretion into the synapse. Especially since it is known that egh and brn mutants show enlarged nerves (Dahlgaard et al., 2012). There are no representative images for MacCer and Fz2 fluorescent intensities in egh mutants and wild types in Figure 6—figure supplement 2C. Again, not all quantified genotypes show representative images in main Figure panel L.

We have added Pearson Coefficient analysis of all colocalization results in Figure 6—figure supplement 1 and quantifications of Western results in Figure 6—Figure supplement 3. We have added representative images for MacCer and Fz2 staining in Figure 6—figure supplement 4 and representative images for Futsch staining of all genotypes quantified in Figure 6—figure supplement 2.

To support the notion that the transport of Wg to synaptic terminals may be disrupted, we have added data showing that the level of Wg-HA in the neuropil of the ventral nerve cord was increased in *egh* mutants in Figure 6—figure supplement 3. In contrast, Wg-HA level in axons within segmental nerves of *egh* mutants was normal. Thus, there may be an axonal transport defect of Wg in *egh* mutants, but this point is beyond the scope of this study.

The enlarged nerves in *egh* mutants are due to over-proliferation of glia and accumulation of immune cells (Dahlgaard et al., 2012). Dahlgaard et al., (2012) does not mention a possible axonal transport defect in *egh* mutants.

*- Figure 8: In silico predictions and physiochemical experiments suggest a direct binding of Wg to MacCer, but there is no evidence that this is also true* in vivo*. The authors made the wg GBM mutant and claim that this would affect lipid rafts via disrupted MacCer binding. However, the presented data are not very direct and the implication of MacCer levels and localization in wg GBM mutant animals is lacking.*

To further support the binding of Wg and MacCer, we have added new pull-down data showing that wild-type Wg binds LacCer whereas Wg carrying mutated GBM does not in Figure 8. We also have added new in vivodata in Figure 9 showing a significantly reduced co-localization between MacCer and Wg in *wg^1^/wg^GBM^* mutant NMJs compared to that in both wild type and *wg^1^*/+ heterozygous control. Although we could not direct examine homozygous *wg^GBM^* mutants which are embryonic lethal, the comparison between *wg^1^*/+ heterozygous and *wg^1^/wg^GBM^* mutant effectively revealed the effect of *wg^GBM^* allele on Wg-MacCer co-localization.

We did not claim that wg GBM mutant would affect lipid rafts via disrupted MacCer binding. Instead, we showed that MacCer level and staining pattern at NMJs in *wg^1^/wg^GBM^* and *wg^1^/wg^CX4^*mutants were not changed compared to that of *wg^1^*/+ heterozygous and wild-type control (Figure 9 and Figure 6—figure supplement 4). Together, our data show that MacCer positively regulates Wg level at NMJs, but not vice versa.

Reviewer #2:

[…] Overall, while the topic is interesting, several data sets are over-interpreted, and controls are missing. The paper tries to make many points, but the data is lacking for some parts of their model. This is particularly true for the claims on SL synthesis (like Figure 1), as well as the concept that MacCer is active in lipid rafts and/ or endosomes. On the other hand, there is considerable data on the point that Egh (MacCer) genetically interacts with Wg and affects presynaptic Wg signalling – though again the manuscript falls short of convincingly proving a direct physical interaction (Figure 8).

We have toned down the statements regarding sphingolipid synthesis and lipid rafts results. We have rephrased the title of sphingolipid synthesis result to “Mutations in de novo sphingolipid synthetic enzymes affect NMJ growth”.

GSLs are critical raft-components and used as molecular markers for membrane lipid rafts (reviewed in Simons and Gerl, 2010). It has been well established that lipid rafts/LacCer (the vertebrate analog of MacCer) are enriched in recycling endosomes (Balasubramanian et al., 2007; Gagescu et al., 2000; Hortsch et al., 2010). These references have added in the revision (also see the response to the last point of this reviewer). Our result that MacCer co-localizes with Syx1A and Rab11 is consistent with the concept that MacCer is part of lipid rafts and enriched in recycling endosomes.

We have carried out a lipid pull-down assay which shows a direct binding between Wg and LacCer (revised Figure 8).

One issue is that the first half of the paper (Figures 1-5) is limited to bouton numbers and bouton sizes. However, many molecular pathways converge to affect these variables and it is not possible to confidently assign genes (or drug treatments) to a single common molecular pathway. The authors should add additional readouts of synaptic development to see if their conclusions on genetic interactions stand up. There may be parts of this in the final figures showing Wg-related defects in Egh lines, but these analyses were not conducted for the majority of genetic interactions shown in Figures 1-5.

We agree that many molecules and pathways converge to affect NMJ growth (reviewed in Harris and Littleton, 2015). To identify which pathway is directly regulated by MacCer, we have examined a few molecules and pathways that are known to regulate NMJ growth. TGF-β/BMP is an important retrograde growth promoting signaling in NMJs. We previously reported that the NMJ growth-promoting effect of MacCer is not dependent on this signaling pathway (Huang et al., 2016). In this revision, we have added new data showing a normal expression of important synapse regulators such as FasII and Syx1A at *egh* mutant and *brn*-overexpression NMJ (Figure 3—figure supplement 1). Instead, our substantial data (Figures 7–9) support that MacCer acts as an adaptor for Wg to facilitate Wg signaling at NMJ synapses.

Related is that there are several experiments that use constitutive hypomorphs/ knock-out lines of widely expressed genes but interpret data to conclude that genes act presynaptically. For example, Figure 1 shows synaptic defects in constitutive mutants of Schlank, lace and Sk2, without genetically rescuing to show that they operate presynaptically. The Figure 5 genetic interaction between Wg and Egh uses many constitutive mutants. The authors need to better establish that all phenotypes are cell autonomous to motor neurons (RNAi and re-expression in mutant backgrounds with motor neuron-specific drivers).

Our new results from motoneuron specific manipulation of gene expression by *OK6-Gal4* support that GSL synthase Egh and Brn regulate NMJ growth in presynaptic neurons but not glia or postsynaptic muscles (Figure 2 and Figure 3).

In addition to the neuronal specific role of Brn, we showed that neuronal co-expression of Brn and Wg driven by *nsyb-Gal4* regulates NMJ growth, suggesting an interaction between MacCer and Wg presynaptically.

This problem is amplified with the pharmacological treatments. Drugs that deplete sterols or GSL from egg hatching likely broadly impair development rather than specifically affect lipids in synapses as the authors interpret. At the very least control data is needed on larval size, time to pupate, and general tissue development, etc. The conclusion that sterol and SL depletion data show MacCer operating in lipid rafts is overinterpreted given the drug treatments will affect all cells in the animal from egg hatching and have multiple impacts at the level of neuronal function (not just disrupting presynaptic lipid rafts).

As far as we know, we are the first to use the two drugs in *Drosophila* larvae. We tried the two drugs in various concentrations. Treatment with a high concentration of MβCD and Filipin led to embryonic lethality. Therefore, we selected a low concentration of drugs that affected NMJ development without apparent larval growth defects. We have added description on larval size, developmental time, and muscle size upon drug treatment in the revision (Figure 4—figure supplement 1). We agree that depletion of sphingolipids or sterols by drug or genetic means is expected to have a general effect on the whole animal, but the effect seems to be specific on NMJ growth based on our assays (Figure 4 and Figure 5 and associated supplements).

The final parts of the paper contain the more convincing data on a genetic interaction between Egh and Wg, including multiple signs of abnormal Wg singling in Egh mutants. However, it seems particularly critical to be sure of direct binding between Wg and MacCer and ideally the authors can show this via a second methodology since the manuscript has too little experimental data to prove this.

As suggested by the reviewer, we have carried out pull-down assay which shows that Myc-Wg binds with LacCer whereas Myc-Wg containing mutated GBM does not (Figure 8G). We also have added new in vivodata in Figure 9 showing a significantly reduced co-localization between MacCer and Wg in *wg^1^/wg^GBM^* mutant NMJs compared to that in both wild type and *wg^1^*/+ heterozygous control. We believe these new results together with our previous membrane lipid binding data in Figure 8 (D–F) support a direct binding between MacCer and Wg via GBM.

The concept that MacSer is specifically important and localised to endosomes also isn't fully established. If important, the authors should biochemically test this, for example via lipid MS of purified endosomes.

We provided experimental evidence that MacCer co-localizes with recycling endosome marker rab11 at the presynaptic NMJ terminals (Figure 6). Localization of lipid rafts/LacCer to recycling endosomes has been firmly established in the literature. For example, fluorescence labeled LacCer is enriched in recycling endosomes in human and *Drosophila* cells (Hortsch et al., 2010). Similarly, recycling endosomes purified from Madin-Darby canine kidney cells are enriched in raft-resident lipids and raft-associated proteins (Gagescu et al., 2000). These two references have been added in subsection “MacCer facilitates local presynaptic Wg signaling at NMJs” and subsection “MacCer promotes synaptic growth via presynaptic Wg signaling” in the revision.

Reviewer #3:

[…] The generation of the GBM Wingless mutant via CRISPR is an excellent new tool for the investigators to further explore the role of Wg-GSL interactions in Wg function. While I acknowledge that extensive characterization of this new allele is beyond the scope of this study, a few more experiments are necessary to test their hypothesis. First, the authors should examine bouton size in Wg[GBM/1] animals as they have done in previous figures to determine if this phenocopies mutants that cause reduction of MacCer. Second, the authors should show that these phenotypes can be rescued by presynaptic expression of wildtype Wg. Finally, it would be of great interest to determine if the Wg[GBM/1] phenotype is modulated by egh and brn genetic manipulations as was performed in prior figures. If the Wg[GBM/1] phenotype is epistatic to brn and egh as would be predicted by the model, an insensitivity of this GBM allele to MacCer levels would be strong genetic evidence that these GSLs are regulating synaptic development through Wg.

As the reviewer suggested, we have added the data showing that *wg^GBM^/wg^1^* mutants show fewer and larger synaptic boutons (Figure 9J and N), recapitulating the phenotype of MacCer deficiency *egh* mutants. TheNMJ undergrowth phenotype in *wg^GBM^/wg^1^* mutants was rescued by presynaptic expression of Wg (Figure 9, K and N). The NMJ phenotype of *wg^GBM^/wg^1^ (wg^1^* is a hypomorph mutation) was insensitive to the increased level of MacCer, supporting that Wg is epistatic to MacCer (Figure 9, L and N).Based on early lethality of *wg^GBM^/wg^GBM^* and *wg^CX4^/wg^GBM^*, and similar NMJ phenotype of *wg^GBM^/wg^1^* and *wg^CX4^/wg^1^* (Figure 5 and Figure 9), we assume that *wg^GBM^* behaves as a genetic null as *wg^cx4^*.

The alteration in MacCer levels in egh mutants are quite modest (Figure 3B). Is this because of the partial LOF nature of the alleles or due to alternative ways of generating MacCer? In this regard, the data provided (Figure 8—figure supplement 1C) don't convince me of their conclusion that "egh62d18 mutants contained chiefly GlcCer ".

*egh* is thought to be a single copy gene without apparent homologues capable of providing functional redundancy based on sequence analysis (Wandall et al., 2003; Wandall et al., 2005). *egh^62d18^* mutation carries an 11 bp deletion which causes a frameshift at amino acid 97 resulting in most of the protein sequence (458-97 amino acids) not expressed; thus, it is presumably a null (Wandall et al., 2005). The change of MacCer intensity was indeed modest (Figure 3B). This is because of non-specific background staining in *egh* nulls, residual MacCer produced in *egh* mutants by an alternative approach, or both. For Figure 8—figure supplement 1C, we rephrased it into “*egh^62d18^* mutants lacked MacCer while *brn^1.6P6^* mutants contained primarily MacCer” in the revision.

In Figure 2L, the authors show that brn[fs107/Y] mutants have a significant increase in bouton no. Presumably, this is compared with WT male larvae, though the control should be explicitly stated. What is the WT strain used, and is it the appropriate genetic background control? In Figure 2, which egh allele is used in the rescue experiments?

*w1118* was used as the wild-type control in this study, as in Materials and methods section. *w1118* is widely used as wild-type control in NMJ studies by many laboratories, including that of Vivian Budnik, Graeme Davis, Barry Ganetzky, Patrik Verstreken (Ballard et al., 2014; Kerr et al., 2014; Khuong et al., 2010; Pielage et al., 2011; Zhao et al., 2015). We used both male and female *w1118* as control because gender does not significantly affect the NMJ growth. The genotype of *egh* rescue experiment in Figure 2 was *egh^62d18^*/Y; *UAS-Egh*/+; *nsyb-Gal4*/+.

[Editors' note: the author responses to the re-review follow.]

1) What is the impact of MacCer localization in lipid rafts on interaction with Wg?

Previous reports show that MacCer (short for mannosylglucosylceramide) and Wg (through its acylation) are enriched in lipid raft domains of membrane by biochemical fractionation (Rietveld et al., 1999; Zhai et al., 2004). Our study revealed a colocalization of MacCer and Wg at presynaptic NMJs (Figure 6A), consistent with direct physical binding of MacCer and Wg (Figure 8). We have added detailed discussion of the impact of lipid raft on MacCer-Wg interaction in the revision in subsection “MacCer promotes synaptic growth via presynaptic Wg signaling”.

2) The authors only detect internal signal with the MacCer antibody to examine lipid rafts at NMJs.

In addition to MacCer staining, we also stained NMJs with an antibody against Syx1A (Figure 4A), a protein marker of lipid rafts (Chamberlain et al., 2001). The Syntaxin 1A puncta substantially overlapped with MacCer puncta with or without colocalization with HRP-positive plasma membrane (Figure 4A), confirming the presence of lipid rafts at NMJs.

3) It feels like an over-interpretation about the genetic results on the basis that the double manipulation is similar to sterol or MacCer loss alone.

In addition to the observation that depleting sterol with MβCD could not exacerbate the NMJ undergrowth in MacCer-deficient larvae, we also showed that MβCD fully restrict the NMJ overgrowth in MacCer-excessive larvae to the level of MβCD-treated wild-type larvae (Figure 4G and Figure 4—figure supplement 1F). We have added these results in the revised text in subsection “MacCer promotes NMJ bouton formation in alipid raft-dependent manner”. We agree that pharmacological approaches may have other effects rather than just affect lipid rafts. Thus, we toned down the conclusion that sterol may modulate NMJ growth in a common genetic pathway with sphingolipids and MacCer.

4) The authors suggest that lipid rafts have some impact on MacCer function, without further elucidating on what this might be.

The restriction of synaptic growth upon lipid rafts disruption may be due to Wg de-localization from lipid rafts rather than a reduction in the synaptic level of MacCer per se. Lipid raft disruption de-localizes Wg from the rafts evidenced by biochemical fractionation (Zhai et al., 2004); delocalization of Wg disrupts its accessibility and binding to MacCer, compromises its transport to the NMJ terminals (Figure 5—figure supplement 1 L-N), and in turn restricts NMJ growth. We have added these arguments in the revision (see subsection “MacCer promotes synaptic growth via presynaptic Wg signaling”).

5) Why does MacCer staining remain punctate even after lipid rafts are disrupted by sterol depletion?

It is reasonable to expect that MacCer would diffuse or appear less punctate upon lipid raft disruption. The microscale MacCer puncta in our assays are not representing single nanoscale lipid raft assemblies. It is thus possible that the formation dynamics of the nanoscale lipid rafts could not be discerned by confocal microscopy, as the nanoscale microdomains are below the optical resolution limit set by the light diffraction (Eggeling et al., 2009; Simons and Gerl, 2010). This possibility is consistent with a previous finding that upon sterol depletion by MβCD, the localization of a raft protein LFA-1 was unaltered examined by confocal microscopy at subcellular level but altered by single-molecule near-field optical microscopy at molecular level (Van Zanten et al., 2009). This point has been added in the revised Results section.

The MacCer antibody is the only readout of MacCer levels (there is no direct analysis of lipid levels in the mutants). It would, therefore, be good if they provide data on antibody specificity. This could include ruling out the possibility that it detects a different molecule that also rises and falls with altered synaptic size; for example, whether MacCer staining remains normal with other manipulations that affect synapse structure (like in Wg mutants). Some of this may be already in the manuscript, but it would help if the issue is directly addressed given non-specific labeling would invalidate many of their conclusions. I also wonder about antibody accessibility when proteins interact with MacCer. Presumably, Wg and the antibody interact with the carbohydrate group of MacCer, so the MacCer antibody would detect less MacCer if it is already complexed with Wg (or other binding partners)? This seems like something to consider when interpreting colocalization data and when using the antibody to determine synaptic MacCer levels.

In addition to MacCer immunostaining, high performance thin layer chromatography (HPTLC) analysis also showed that *egh^62d18^* mutants lacked MacCer and *brn^1.6P6^* mutants contained almost exclusively MacCer (Figure 8—figure supplement 1; Hamel et al., 2010; Wandall et al., 2005). A few mutants with altered NMJ morphology show normal synaptic MacCer levels, including *desat1* (Huang et al., 2016), *rab11, wg^1^/wg^CX4^* (Figure 6*—*figure supplement 4 A-D) and *wg^1^/wg^GBM^* (Figure 9 A-D) mutants. Based on these staining results of mutants and overexpressing linesof *egh* and *brn*, we are confident that the anti-MacCer staining is specific. We have added related description in subsection “GSL MacCer promotes NMJ growth” of the revision.

It is a legitimate question whether the binding partners of MacCer would affect the antibody accessibility to MacCer. Based on our results, Wg does not affect MacCer immunoreactivity, as *wg* mutation does not alter MacCer intensity at NMJ (Figure 6*—*figure supplement 4 A-D andFigure 9A-D).